# Polynucleotide phosphorylase protects against renal tubular injury via blocking mt-dsRNA-PKR-eIF2α axis

Yujie Zhu[1,4], Mingchao Zhang[1,2,4], Weiran Wang[1,4], Shuang Qu[3,4], Minghui Liu[3], Weiwei Rong[1], Wenwen Yang[1], Hongwei Liang[3], Caihong Zeng[2], Xiaodong Zhu[2], Limin Li [3] ✉, Zhihong Liu [2] ✉ & Ke Zen [1,3] ✉

Renal tubular atrophy is a hallmark of chronic kidney disease. The cause of tubular atrophy, however, remains elusive. Here we report that reduction of renal tubular cell polynucleotide phosphorylase (PNPT1) causes renal tubular translation arrest and atrophy. Analysis of tubular atrophic tissues from renal dysfunction patients and male mice with ischemia-reperfusion injuries (IRI) or unilateral ureteral obstruction (UUO) treatment shows that renal tubular PNPT1 is markedly downregulated under atrophic conditions. PNPT1 reduction leads to leakage of mitochondrial double-stranded RNA (mt-dsRNA) into the cytoplasm where it activates protein kinase R (PKR), followed by phosphorylation of eukaryotic initiation factor 2α (eIF2α) and protein translational termination. Increasing renal PNPT1 expression or inhibiting PKR activity largely rescues IRI- or UUO-induced mouse renal tubular injury. Moreover, tubular-specific PNPT1-knockout mice display Fanconi syndrome-like phenotypes with impaired reabsorption and significant renal tubular injury. Our results reveal that PNPT1 protects renal tubules by blocking the mt-dsRNA-PKR-eIF2α axis.

Chronic kidney disease (CKD), defined by persistent albuminuria and/or decline in glomerular filtration rate (GFR), is a serious disease that affects 1 in every 7-10 adults worldwide[1]. Although the research about CKD has long been focused on glomerulus, accumulating evidences has demonstrated that renal tubular injury and atrophy is also a hallmark of CKD[2-5]. In fact, the extent of renal dysfunction of CKD patients correlates more closely with tubular injuries, including tubular atrophy and interstitial fibrosis, than with changes in glomerular morphology[6]. Given the significance of tubular atrophy in CKD progression and as a predictor of GFR decline superior to glomerular nephropathy[7], various investigators have studied the causes of renal tubular atrophy, including tubular epithelial cell apoptosis, cell senescence, peritubular capillary rarefaction and downstream tubule ischemia, oxidative stress, epithelial-to-mesenchymal transition and interstitial

inflammation[8]; however, the mechanisms leading to renal tubular atrophy in CKD are still poorly understood.

Renal tubular cells, accounting for approximately 90% of the cortex, are highly enriched in mitochondria[9]. The primary function of the proximal tubules is reabsorption, and such active process requires a large amount of energy produced by mitochondria[10]. It has been widely reported that mitochondrial dysfunction, including mitochondrial biogenesis, dynamics, and oxidative stress, is tightly linked to renal tubular cell disease[11,12]. He et al. identified that a decreased supply of fuel and increased demand for oxygen by mitochondria in renal tubules are pivotal pathological mechanisms of diabetic tubulopathy in diabetic kidney disease (DKD)[11]. Renal tubular dysfunctional mitochondria may accelerate the initiation of early-stage kidney diseases such as diabetic tubulopathy[12]. Mitochondrial dysfunction is also inseparable from

[1]State Key Laboratory of Pharmaceutical Biotechnology, Jiangsu Engineering Research Center for MicroRNA Biology and Biotechnology, Nanjing University School of Life Sciences, Nanjing, Jiangsu 210046, China. [2]National Clinical Research Center of Kidney Diseases, Jinling Hospital, Nanjing University School of Medicine, Nanjing, Jiangsu 210002, China. [3]School of Life Science and Technology, China Pharmaceutical University, Nanjing, Jiangsu, China. [4]These authors contributed equally: Yujie Zhu, Mingchao Zhang, Weiran Wang, Shuang Qu. ✉e-mail: liminli@cpu.edu.cn; liuzhihong@nju.edu.cn; kzen@nju.edu.cn

increased reactive oxygen species production and energy expenditure by tubular cells, while the glycolytic pathway may serve as a self-protection mechanism in DKD[13]. To meet aerobic metabolism and high energy requirements, mitochondria-rich renal proximal tubule cells must maintain mitochondrial homeostasis and dynamic turnover, which makes renal proximal tubule cells very vulnerable to mitochondrial impairment[9]. The homeostasis and dynamic turnover of mitochondrial double-stranded RNAs (mt-dsRNAs) are also critical to cell fate. To most mammalian cells, cytosolic dsRNAs remain an essential threat[14]. Increased oxidative stress in tubular cells under CKD conditions not only increases the generation of mt-dsRNAs, but also results in their leakage from the mitochondria into the cytoplasm. Recently, numerous studies reported the involvement of mt-dsRNAs efflux into cytoplasm to activate the innate immune system in various human diseases, including alcohol-induced liver damage[15], Huntington's disease[16], and osteoarthritis[17]. In the cytoplasm, various RNA sensor proteins can recognize these mt-dsRNAs[18–20]. One such sensor protein is dsRNA-activating protein kinase R (PKR), which can further activate eukaryotic initiation factor 2α (eIF2α) at Ser51, leading to termination of general protein synthesis[18,21,22]. However, the cellular relocation of mt-dsRNAs in renal tubular cells under various disease condition and their potential function in renal atrophy have not been examined.

To explore the potential role of mt-dsRNAs in promoting renal tubular injury, we undertook an unbiased approach by examining kidney tissues from patients with different renal dysfunctions, as well as from mice with ischemia reperfusion-induced injury (IRI) or unilateral ureteral obstruction (UUO) procedures. These tissues all displayed renal tubular injury in various degrees compared to their respective controls. We found that, in injured renal tubular cells, mt-dsRNAs were widely spread in the cytoplasm where they activated the PKR-eIF2α signaling axis, resulting in termination of general protein synthesis, cell injury and apoptosis. This study further identified the reduction of renal tubular PNPT1 induced by TGFβ1, hyperglycemia or LPS as the mechanism underlying the relocation of mt-dsRNAs from mitochondria into the cytoplasm. Supporting the role of PNPT1 in controlling renal tubular cell mt-dsRNA homeostasis and injury, renal tubular-specific PNPT1-deficient mice exhibited severe impaired reabsorption and significant renal tubular injuries. The present study reveals an important role of PNPT1 in protecting renal tubules via inhibiting the mt-dsRNA-PKR-eIF2α signaling axis.

## Results

### Detection of cytosolic dsRNAs in atrophic renal tubular cells but not podocytes

Given that cytosolic dsRNAs are an essential threat for all mammalian cells[14], we examined the level and location of dsRNAs in the cytoplasm of various renal cells under normal or disease conditions using J2 antibody, which specifically stains dsRNAs. Kidney tissue sections from patients with ongoing acute tubular necrosis (ATN) who displayed severe renal tubular injury were analyzed. As shown in Fig. 1a, a significant amount of J2-labeled dsRNAs was detected in renal tubular cells from ATN patients but not in non-injured renal tubular cells derived from the paracancerous kidney tissues. Interestingly, no dsRNA was detected in podocytes from ATN patients and non-renal tubular injury donors, suggesting a potentially specific linkage between cytosolic dsRNAs and renal tubular injuries. Quantitative analysis of cytosolic dsRNAs showed a strong increase of cytosolic dsRNA level in injured renal tubules from ATN patients ($n = 12$) compared to renal tubules from non-renal tubular injury donors ($n = 5$) (Fig. 1b). To confirm the high level of cytosolic dsRNAs in renal tubular cells as the common cause for renal tubular injury in various kidney diseases, we examined kidney tissue sections from patients with various renal dysfunctions, including diabetic nephropathy (DN, $n = 14$), lupus nephritis (LN, $n = 15$), IgA nephropathy (IgAN, $n = 16$), membranous nephropathy (MN, $n = 13$) or focal segmental glomerulosclerosis

(FSGS, $n = 15$). Various degrees of renal tubular injury in all these tissues were confirmed pathologically. As shown in Fig. 1c, aquaporin 1 (AQP1)-expressing renal tubules (green) from various kidney diseases all exhibited significant damage compared to control renal tubules. Correlatively, a large amount of J2-stained dsRNA (red) was detected in the cytoplasm of injured tubular cells. Quantitative analysis further showed that cytoplasmic dsRNA levels in renal tubular cells were positively correlated to the degree of tubular injury (Fig. 1d).

IRI and UUO procedures are two experimental models widely used to study renal tubular injury and fibrosis[23–30]. To characterize the re-location of dsRNA and its role in renal tubular injury, we established experimental renal tubular injury mouse models using IRI (Supplementary Fig. 1) and UUO procedures (Supplementary Fig. 2). As shown, IRI and UUO procedures caused significant mouse renal tubular injuries, as indicated by the serum creatinine (Scr) levels (Supplementary Figs. 1b and 2b), urinary levels of renal tubular injury markers including KIM-1, β-MG and NGAL (Supplementary Figs. 1c and 2c) and higher renal tubular injury score (Supplementary Figs. 1d and 2d). We next detected cytosolic dsRNAs in mouse renal tubular cells following either IRI or UUO procedures (Fig. 1e). Examination of mouse renal tubular injury and cytosolic dsRNA level in renal tubular cells demonstrated a positive correlation between renal tubular injury induced by IRI or UUO and cytosolic dsRNA levels (Fig. 1f).

Previous study reported that mitochondrial dsRNAs (mt-dsRNAs) could leak into cytoplasm where they trigger antiviral signaling in humans[31]. We next examined the potential linkage between cytosolic mt-dsRNAs and renal tubular injury. To determine the source of dsRNAs in the injured renal tubules, we purified dsRNAs from mouse proximal renal tubules using a J2 antibody. In this experiment, renal tubules were isolated from mice with or without IRI. Mitochondria-free cytosolic fraction was further isolated as depicted in Supplementary Fig. 3. RNA-seq analysis of renal tubular cytosolic dsRNAs indicated a panel of mt-dsRNAs, including *mt-ND4-6*, *mt-CO1* and *mt-CYB*, in injured mouse renal tubules but not in normal renal tubules. Employing the specific probes against these mt-dsRNAs, we quantified their cytosolic levels in proximal renal tubules of mice with or without IRI (Fig. 1g). Compared to Sham group that showed almost no cytosolic mt-dsRNA, mice with IRI displayed significant amount of *mt-ND4-6*, *mt-CO1* and *mt-CYB* in renal proximal tubular cytoplasm. Strongly increase of cytosolic mt-dsRNA level in injured mouse proximal renal tubules was confirmed by using specific probes against heavy and light strand of these mitochondrial RNAs (Fig. 1h).

### Increase of cytosolic dsRNA level in atrophic renal tubule is due to PNPT1 reduction

Given that homeostasis of mitochondrial RNAs in mammalian cells is controlled by PNPT1, a polynucleotide phosphorylase that forms the degradosome with SUV3 helicase to degrade RNA[32,33], we explored whether the relocation of mitochondrial dsRNAs (mt-dsRNAs) into the cytoplasm in atrophic renal tubular cells is caused by loss of PNPT1 function. In agreement with the notion that PNPT1 is a polynucleotide phosphorylase mainly associated with the mitochondria[34–36], we detected a much higher level of PNPT1 in renal tubules than in glomeruli, as renal tubular cell possesses significantly more mitochondria than a podocyte (Supplementary Fig. 4). Compared to non-renal tubular injury donors, patients with various renal dysfunctions displayed a markedly reduced PNPT1 levels in injured renal tubules (Fig. 2a). Quantification of PNPT1 level and tubular injury degrees further indicated that renal tubular PNPT1 levels are negatively correlated to the degree of renal tubular injury (Fig. 2b). Reduction of renal tubular PNPT1 was also observed in mice with IRI and UUO procedures. Both immunostaining of mouse kidney tissue sections (Fig. 2c, d) and WB analysis of isolated mouse tubules (Fig. 2e) showed that mouse renal tubular PNPT1 is markedly downregulated during renal tubular injuries induced by either IRI or UUO procedure.

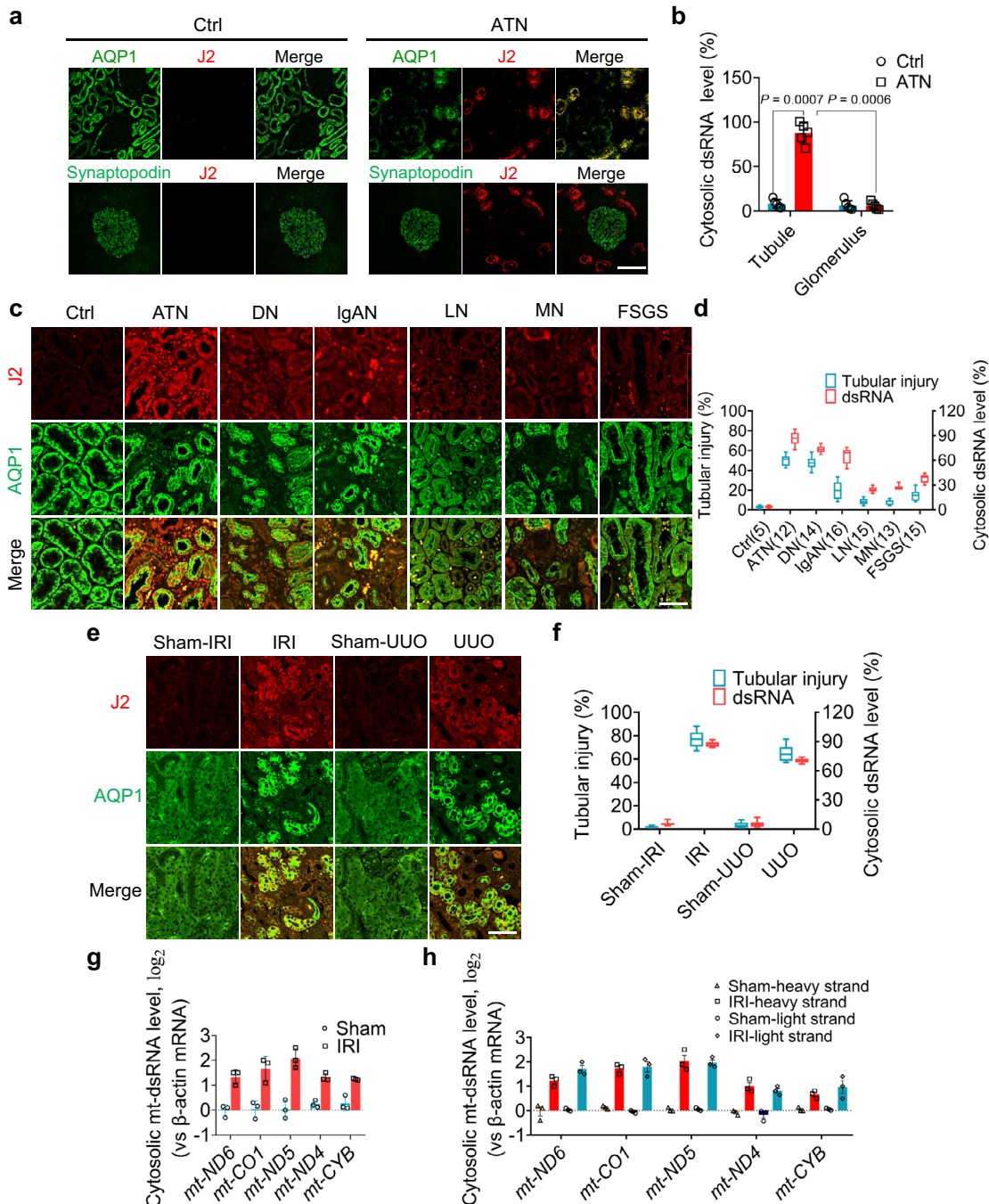

**Fig. 1 | Detection of cytosolic mt-dsRNAs in injured renal tubular cells.**
**a** Immunostaining of cytosolic dsRNA by J2 antibody in tubule and glomerulus from acute tubular necrosis (ATN) patients and controls. Tissue sections were stained with anti-aquaporin 1 (AQP1) antibody (tubular cell) or anti-synaptopodin antibody (podocyte). **b** Quantification of cytosolic dsRNA levels in tubules and glomerulus (5 patients/group). **c** Immunostaining of dsRNA (J2) from 85 patients with various degrees of renal tubular injury, including ATN (5 males and 7 females), diabetic nephropathy (DN, 8 males and 6 females), IgA nephropathy (IgAN, 9 males and 7 females), lupus nephritis (LN, 5 males and 10 females), membranous nephropathy (MN, 8 males and 5 females) and focal segmental glomerulosclerosis (FSGS, 6 males and 9 females), as well as 5 non-renal tubular injury controls (3 males and 2 females). **d** Evaluation of tubular injury degree and cytosolic dsRNA levels.

**e** Cytosolic dsRNA staining by J2 antibody in renal tubule from ischemia reperfusion-induced injury (IRI) or unilateral ureteral obstruction (UUO) mouse models. **f** Evaluation of renal tubular injury degree and cytosolic dsRNA levels in IRI or UUO mouse models. **g** Cytosolic mt-dsRNA levels in renal tubules of IRI and Sham mouse groups. **h** Renal tubular cytosolic levels of mt-dsRNAs detected using specific probes against the heavy and light strand (5 mice/group). Scale bars, 50 μm. The above experiments were successfully repeated three times. Two-way ANOVA with Sidak's multiple comparisons test was performed in **b**, and the results were presented as mean ± SEM. In box plots (**g**, **f**), the centre line shows the median, lower and upper hinges of boxes represent 25th to 75th percentiles, and whiskers extend to minimum and maximum values. Source data are provided as a Source Data file.

## Reduction of renal tubular cell PNPT1 by HG, TGFβ or LPS treatment leads to relocation of mt-dsRNAs into cytoplasm

Previous studies have suggested that PNPT1 expression can be modulated by viral infection-induced interferon (IFN)[37]. To validate PNPT1 reduction in injured renal tubular cells, we treated renal tubular cells with a high concentration of glucose (HG), TGFβ1 or LPS to cause cell damage[38,39]. In this experiment, HK2 cells were treated with 40 mM glucose for 7 days, TGFβ1 (10 ng/mL) for 48 h or LPS (75 μg/mL) for

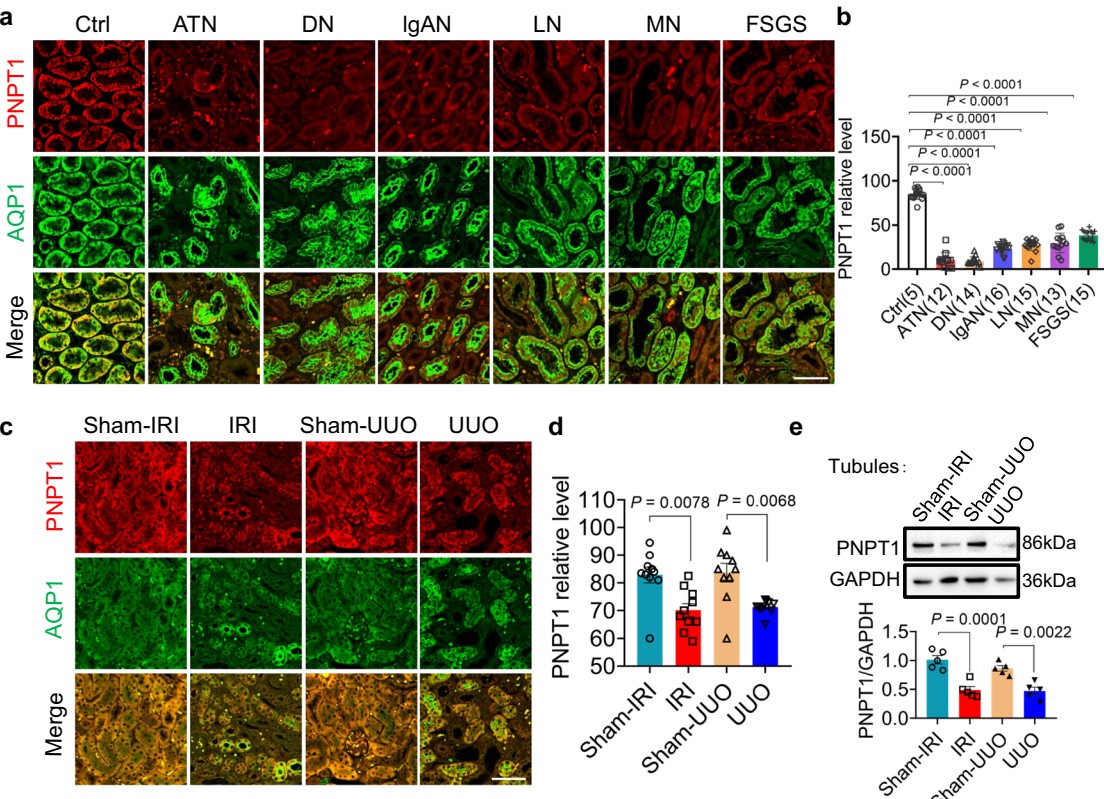

**Fig. 2 | Reduction of PNPT1 in injured renal tubule. a** Immunostaining of PNPT1 (red) in kidney sections from patients with various degrees of renal tubular injury or from non-renal tubular injury controls as previously described in Fig. 1. Renal tubules were stained with anti-AQP1 antibody (green). **b** Quantification of renal tubular PNPT1 level in patients and controls. Renal tissue samples were from 85 patients including ATN (5 males and 7 females), DN (8 males and 6 females), IgAN (9 males and 7 females), LN (5 males and 10 females), MN (8 males and 5 females) and FSGS (6 males and 9 females), as well as 5 non-renal tubular injury controls (3 males and 2 females). Images of 10 randomly selected regions per sample were obtained and analyzed. **c** Immunostaining of PNPT1 (red) in kidney sections from mice with or without IRI or UUO procedure. **d** Quantification of renal tubular PNPT1 levels in mice with or without IRI or UUO procedure (10 mice/group). **e** Top: Western blot (WB) of renal tubular PNPT1 in mice with or without IRI or UUO procedure. Bottom: quantification of protein levels from 3 independent WB experiments. Scale bars, 50 μm. The above experiments were successfully repeated three times. One-way ANOVA with Dunnett's multiple comparisons test was performed in **b**, Tukey's multiple comparisons test was performed in **d**, **e**, and the results were presented as mean ± SEM. Source data are provided as a Source Data file.

24 h. PNPT1 and cytosolic dsRNAs were labeled with anti-PNPT1 antibody and J2 antibody, respectively, and nuclei were stained with DAPI (blue). As shown in Fig. 3a, in HK2 cells injured by HG, TGFβ1 or LPS treatments, cytosolic dsRNA levels were significantly increased while PNPT1 levels were markedly decreased. To explore whether the reduction of PNPT1 is the cause of cytosolic dsRNA increase, we next overexpressed PNPT1 in HK2 cells via lentivirus-mediated gene expression and then monitored the cytosolic dsRNA levels in HK2 cells with the same TGFβ1 treatment. As shown in Fig. 3b, overexpression of PNPT1 not only reversed the reduction of PNPT1 in HK2 cells induced by TGFβ1 treatment, but also completely abolished the elevation of cytosolic dsRNA levels. In contrast, when we directly silenced PNPT1 in HK2 cells using PNPT1-specific siRNA (siPNPT1), we found that, compared to HK2 cells transfected with control oligonucleotide (siCtrl), HK2 cells transfected with siPNPT1 displayed a markedly reduced PNPT1 expression but a significantly higher level of cytosolic dsRNAs (Fig. 3c). These results collectively suggest that reduction of PNPT1 under various injury conditions causes the accumulation of cytosolic dsRNAs in renal tubular cells. Employing the specific probes against *mt-ND4-6*, *mt-CO1* or *mt-CYB*, we also quantified their cytosolic levels in HK2 cells with or without PNPT1 knockdown (Fig. 3d). Compared to control HK2 cells that had almost no cytosolic mt-dsRNA detected, HK2 cells injured by direct PNPT1 knockdown displayed considerable amounts of mt-dsRNAs such as *mt-ND4-6*, *mt-CO1* and *mt-CYB*. Strongly increase of cytosolic levels of *mt-ND4-6*, *mt-CO1* and *mt-CYB* in HK2 cells with PNPT1 knockdown was confirmed by using specific probes against heavy and light strand of these mitochondrial RNAs[17] (Fig. 3e). To directly show the leakage of mt-dsRNA into cytoplasm during tubular cell injury, we performed in situ staining of *mt-ND5* heavy (Fig. 3f) and light strand (Fig. 3g) in HK2 cells with or without PNPT1 knockdown. Cytosolic levels of both heavy and light strand of *mt-ND5* were strongly increased in HK2 cells with PNPT1 knockdown. Furthermore, different from ATP5A1, a mitochondrial marker[40], a considerable amount of heavy and light strand of *mt-ND5* was observed outside mitochondria in HK2 cells transfected with PNPT1 siRNA.

## Role of the mt-dsRNA-activated PKR-eIF2α signaling axis in promoting renal tubular cell injury and apoptosis

As an essential threat to most mammalian cells, cytosolic mt-dsRNAs can result in translational termination by binding and activating PKR, which in turn, phosphorylates eIF2α[41,42], or initiates anti-viral immunity by ligating RIG-I and MDA5 helicases[20,43–45]. To determine the signaling downstream of cytosolic mt-dsRNAs in inducing renal tubular cell injury, we examined the protein synthesis capacity in HK2 cells with or without siPNPT1 plasmid transfection using Click-iT® Plus OPP Protein Synthesis Assay Kit. As shown in Fig. 4a, knockdown of PNPT1 in HK2 cells strongly inhibited whole protein translation at 24 h post-infection, suggesting that the role of cytosolic mt-dsRNAs induced by PNPT1 reduction in HK2 cells is largely through activating PKR to shut off general protein synthesis. As the termination of protein synthesis would eventually lead to cell apoptosis, we examined the HK2 cell survival with or without PNPT1 knockdown, and found that PNPT1

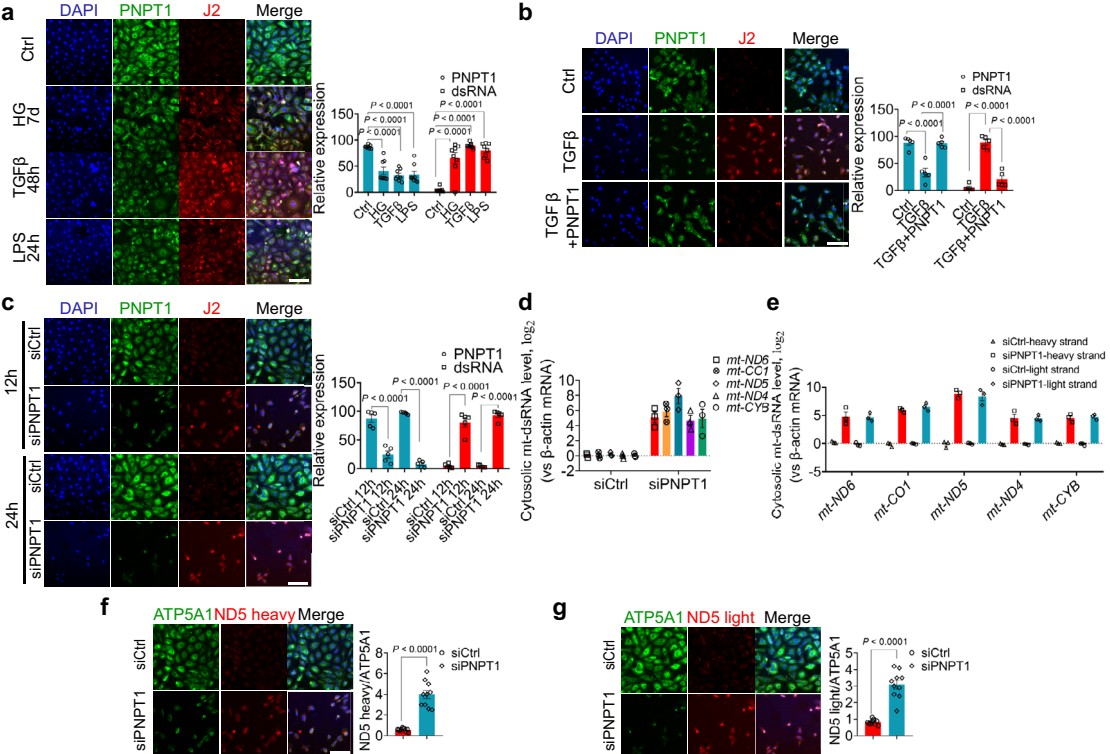

**Fig. 3 | Reduction of PNPT1 in human renal tubular HK2 cells caused relocation of mt-dsRNAs into cytoplasm. a** Left: immunostaining of PNPT1 (green) and cytosolic dsRNA (J2, red) in HK2 cells treated with or without HG (40 mM, 7d), TGFβ1 (10 ng/mL, 48 h) or LPS (75 μg/mL, 24 h). Nuclei were stained with DAPI (blue). Right: quantification of PNPT1 and cytosolic dsRNA levels in HK2 cells. **b** Left: immunostaining of PNPT1 (green) and cytosolic dsRNA (J2, red) in HK2 cells treated with or without TGFβ1 (10 ng/mL, 48 h). Prior to TGFβ1 treatment, HK2 cells were infected with or without PNPT1-expressing lentivirus. Right: quantification of PNPT1 levels and cytosolic dsRNA in HK2 cells. **c** Left: immunostaining of dsRNA (J2, red) and PNPT1 (green) in HK2 cells transfected with PNPT1 siRNA-expressing plasmid (siPNPT1) or control oligonucleotide-expressing plasmid (siCtrl). Right:

quantification of PNPT1 and cytosolic dsRNA levels in HK2 cells. **d** RT−qPCR analysis of cytosolic mt-dsRNAs in HK2 cells transfected with siPNPT1 or siCtrl plasmid. **e** Cytosolic levels of mt-dsRNAs detected using specific probes against their heavy and light strand in HK2 cells transfected with or without siPNPT1 plasmid. In situ staining of cytosolic mt-ND5 heavy (**f**) and light strand (**g**) in HK2 cells transfected with siPNPT1 or siCtrl plasmid. ATP5A1 served as a mitochondria marker. Scale bars, 50 μm. The above experiments were successfully repeated three times. Two-way ANOVA with Sidak's multiple comparisons test was performed in **a**−**c**. Two-tailed unpaired *t* test was performed for the statistical analyses in **f**−**g**, and the results were presented as the mean ± SEM. Source data are provided as a Source Data file.

knockdown enhances cell apoptosis in a time-dependent manner (Fig. 4b). To confirm the activation of PKR and subsequent phosphorylation of eIF2α, we determined the levels of PNPT1, PKR, phosphorylated PKR (p-PKR), eIF2α, phosphorylated eIF2α (p-eIF2α) and activating transcription factor 4 (ATF4), a transcription factor preferentially translated upon eIF2α phosphorylation, in HK2 cells with or without PNPT1 knockdown (Fig. 4c). At 12 and 24 h post-infection, HK2 cells with PNPT1 knockdown displayed significantly higher levels of p-PKR and p-eIF2α compared to control HK2 cells although the total amounts of PKR and eIF2α in PNPT1-knockdown and control HK2 cells remained similar. Translation of ATF4 was markedly enhanced in HK2 cells after PNPT1 knockdown. Supporting the role of PKR activation by mt-dsRNAs in blocking renal tubular cell protein synthesis via phosphorylating eIF2α, we further treated siPNPT1 HK2 cells with PKR inhibitor C16[46] and monitored the levels of PKR, p-PKR, eIF2α, p-eIF2α and ATF4. As shown in Fig. 4d, C16 treatment strongly decreased p-PKR and p-eIF2α levels in HK2 cells with PNPT1 knockdown but did not affect the levels of PKR and eIF2α. In line with this, C16 treatment also markedly reduced the HK2 cell apoptosis induced by PNPT1 knockdown (Fig. 4e). Based on these results, we developed a working model of renal tubular cell injury induced by cytosolic mt-dsRNAs (Fig. 4f). In this model, PNPT1 reduction induced by various injury factors such as TGFβ1, LPS and HG leads to leakage of mt-dsRNA into the cytoplasm where they activate PKR, which in turn, phosphorylates eIF2α, resulting in the termination of protein translation.

Next we explored the protective role of PNPT1 against renal tubular injury in IRI and UUO experimental mouse models. As shown in Fig. 5a, top, adenovirus-expressing PNPT1 (PNPT1 AAV) or control adenovirus (Ctrl AAV) (virus concentration $1 \times 10^{13}$ VG/ml) were injected into mouse kidney (20 μl per mouse) during IRI model establishment. Both WB analysis and immunofluorescence labeling showed an increased PNPT1 expression in renal tubules after PNPT1 AAV injection (Supplementary Fig. 5). WB analysis of mouse renal tubular PNPT1 level confirmed that PNPT1 AAV injection completely reversed PNPT1 downregulation induced by IRI (Fig. 5a, bottom). J2 antibody staining of mouse kidney tissue sections indicated that PNPT1 AAV treatment strongly abolished the elevation of cytosolic dsRNA induced by IRI (Fig. 5b). Compared to Ctrl AAV, which showed no protection on renal tubular injuries induced by IRI, administration with PNPT1 AAV strongly reduced Scr (Fig. 5c, top) and urinary KIM-1 levels (Fig. 5c, bottom) in the IRI mouse model. TUNEL assay of mouse kidney tissue sections further demonstrated that PNPT1 AAV treatment markedly reduced renal tubular apoptosis in IRI mice (Fig. 5d).

A similar approach of PNPT1 or Ctrl AAV injection was employed in the UUO mouse model (Fig. 5e, top). WB analysis of renal tubular PNPT1 level also indicated the opposite of PNPT1 reduction by PNPT1 AAV treatment in UUO mice (Fig. 5e, bottom). Cytosolic dsRNAs staining by J2 antibody in mouse kidney tissue sections also indicated that PNPT1 AAV treatment completely abolished the elevation of cytosolic dsRNA in UUO mice (Fig. 5f). Compared to Ctrl AAV, which

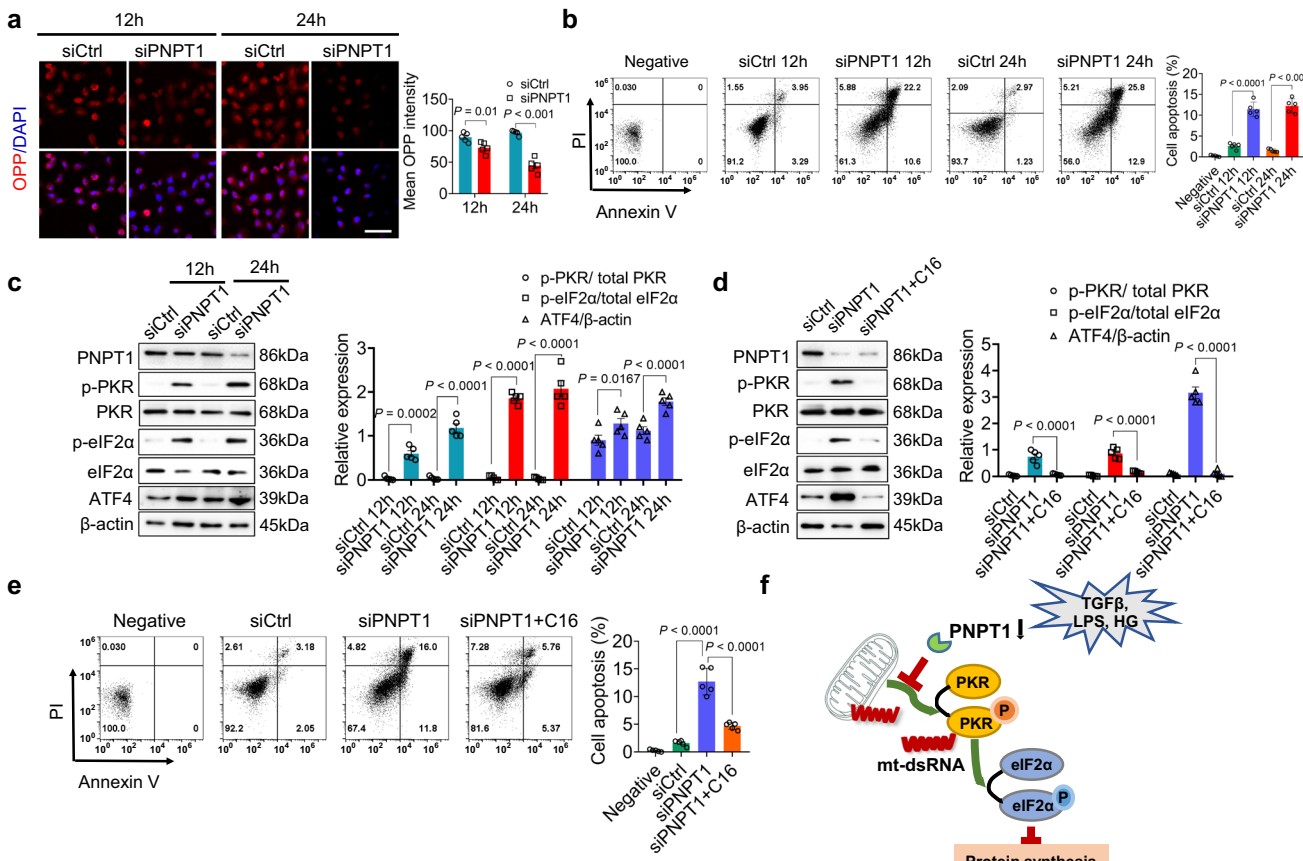

**Fig. 4 | PNPT1 suppressed mt-dsRNA-mediated PKR-eIF2α signaling to prevent renal tubular injury. a** Left: measurement of total protein synthesis in HK2 cells transfected with siPNPT1 or siCtrl plasmid using Click-iT® Plus OPP Protein Synthesis Assay Kit. Right: quantification of OPP protein synthesis. **b** Time course of apoptosis of HK2 cells transfected with siPNPT1 or siCtrl plasmid. **c** Left: Western blot (WB) of p-PKR, p-eIF2α and ATF4 in HK2 cells infected with siPNPT1 or siCtrl plasmid. Right: quantification of protein levels. **d** Left: WB of p-PKR, p-eIF2α and ATF4 in HK2 cells with or without siPNPT1 lentivirus or PKR inhibitor C16. Right:

quantification of protein levels. **e** Apoptosis of siPNPT1-transfected HK2 cells with or without PKR inhibitor C16 treatment. **f** Working model of renal tubular injury induced by mt-dsRNA-PKR-eIF2α axis. Scale bar, 50 µm. The above experiments were successfully repeated three times. One-way ANOVA with Tukey's multiple comparisons test was performed in **a**, **c**, **d**. Two-way ANOVA with Tukey's multiple comparisons test was performed in **b**, **e**. Results were presented as mean ± SEM. Image in **f** was created with BioRender.com. Source data are provided as a Source Data file.

showed no protection against renal tubular injuries in UUO mice, administration with PNPT1 AAV strongly reduced Scr (Fig. 5g, left) and urinary KIM-1 levels (Fig. 5g, right) in UUO mice. Moreover, TUNEL assay of mouse kidney tissue sections confirmed that PNPT1 AAV treatment markedly reduced renal tubular apoptosis in UUO mice (Fig. 5h).

Given the critical role of dsRNA-activated PKR in renal tubular injuries, we next examined the effects of PKR inhibitor C16 on renal tubular damages in mouse models with IRI or UUO. As depicted in Fig. 6a, f, mice were injected intraperitoneally with 500 µg/ml C16 (with DMSO as vehicle control) three days in a row following IRI and UUO procedures, respectively. WB analysis showed that C16 treatment markedly abated renal tubular p-PKR levels in mice with IRI (Fig. 6b). Inhibition of PKR activity by C16 also resulted in decrease of Scr (Fig. 6c, left) and urinary KIM-1 levels (Fig. 6c, right) in mice with IRI. Supporting the protective role of C16 against renal tubular injury, H&E staining of mouse kidney tissue sections revealed that IRI mice treated with C16 exhibited much fewer renal tubular injuries compared to the vehicle group (Fig. 6d), and the TUNEL assay of mouse kidney tissue sections confirmed significantly less renal tubular apoptosis in IRI mice treated with C16 (Fig. 6e). The protective function of C16 was further validated in the UUO mouse model. As shown, C16 treatment markedly decreased renal tubular p-PKR levels (Fig. 6g), Scr (Fig. 6h, left) and urinary KIM-1 levels (Fig. 6h, right) in UUO mice. H&E staining (Fig. 6i)

and TUNEL assay (Fig. 6j) of mouse kidney tissue sections demonstrated that compared to UUO mice treated with DMSO (vehicle group), UUO mice treated with C16 displayed markedly less renal tubular injury and apoptosis. In line with this, C16 treatment largely rescued the protein synthesis in HK2 cells with PNPT1 knockdown, a similar effect achieved by PKR siRNA (Supplementary Fig. 6). Apoptosis of PNPT1-deficient HK2 cells was also significantly decreased by C16 treatment. These results collectively suggest that inhibition of PKR activity can protect against mouse renal tubular injury and apoptosis.

To validate the pathogenic role of cytosolic mt-dsRNAs, we also suppressed the production of mt-dsRNAs using inositol 4-methyltransferase (IMT1), a small molecule inhibitor recently reported by Bonekamp et al.[47]. As shown in Supplementary Fig. 7, IMT1 treatment markedly attenuated renal tubular damage induced by IRI and UUO procedure.

### Renal tubular-specific PNPT1-knockout mice display impaired reabsorption and renal tubular injury

To confirm the role of PNPT1 in controlling mt-dsRNA homeostasis and renal tubular injury, we generated renal tubular-specific PNPT1-knockout mice (KO) using Cre/LoxP system. As shown in Supplementary Fig. 8, the GGT-Cre or floxed only genetic modification had no effect on mouse renal tubular morphology and PNPT1 expression. Genotyping, immunofluorescence staining and WB analysis confirmed

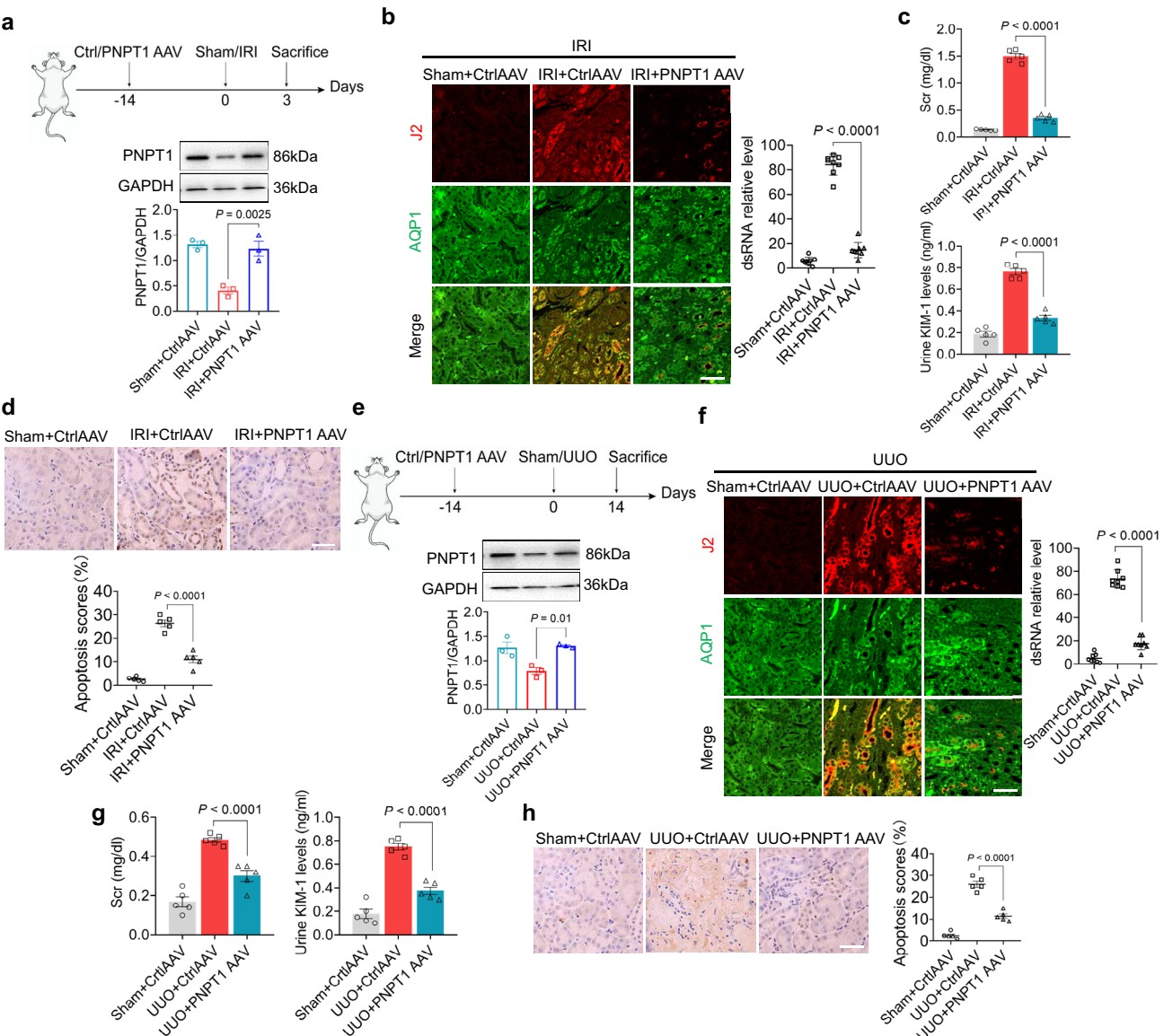

**Fig. 5 | Increasing PNPT1 expression in mouse renal tubule attenuated tubular injury. a** Top, schematic experimental approach of PNPT1-AAV infection in IRI mouse model. Bottom: WB analysis of renal tubular PNPT1 level (3 mice/group). **b** Left: renal tubular dsRNA (J2, red) staining in IRI mouse model with or without PNPT1 AAV infection. Right: quantification of renal tubular dsRNA level (8 mice/group). **c** Serum creatinine (top) and urinary KIM-1 levels (bottom) in IRI mouse model with or without PNPT1 AAV infection (5 mice/group). **d** Top: TUNEL assay of renal tubular apoptosis in IRI model mouse with or without PNPT1 AAV infection. Bottom: quantification of TUNEL assay (5 mice/group). **e** Top: schematic experimental approach of PNPT1-AAV infection in UUO mouse model. Bottom: WB analysis of renal tubular PNPT1 level (3 mice/group). **f** Left: renal tubular dsRNA (J2, red) staining in UUO mouse model with or without PNPT1 AAV infection. Right: quantification of renal tubular dsRNA level (8 mice/group). **g** Serum creatinine (left) and urinary KIM-1 levels (right) in UUO mouse model with or without PNPT1 AAV infection (5 mice/group). **h** Left: TUNEL assay of renal tubular apoptosis in UUO model mouse with or without PNPT1 AAV infection. Right: quantification of TUNEL assay (5 mice/group). Scale bars, 50 μm. The above experiments were successfully repeated three times. One-way ANOVA with Tukey's multiple comparisons test was performed in **a**–**h** and the results were presented as mean ± SEM. Images of mouse in **a**, **e** were created with BioRender.com. Source data are provided as a Source Data file.

the specific knockout of PNPT1 in renal tubules in PNPT1-knockout mice (Fig. 7a). To our surprise, these PNPT1-knockout mice developed a significantly smaller body compared to their WT littermates (Fig. 7b, c). Kidney tissue examination by H&E staining also showed that PNPT1-knockout mice had markedly higher renal injury scores than did WT mice (Fig. 7d). These results suggest that lower body weight in PNPT1-knockout mice is due to renal tubular dysfunction. Supporting this notion, we found that PNPT1-knockout mice exhibited a Fanconi syndrome-like phenotype in terms of impaired reabsorption. Bone mineral density scanning showed that PNPT1-knockout mice had a severe bone structure deformation similar to that in rickets with a

markedly lower bone mineral density compared to WT mice (Fig. 7e). In line with this, significant loss of serum calcium, as well as serum phosphorus at late stage, was found in PNPT1-knockout mice (Fig. 7f).

Renal tubular dysfunction in PNPT1-knockout mice was further analyzed. At age of 8 weeks, PNPT1-knockout mice showed higher levels of glucose, uric acid, phosphorus and potassium in urine compared to WT mice (Fig. 7g). A lower kidney/body weight ratio (Fig. 7h) but higher Scr (Fig. 7i, left) and urinary levels of renal tubular injury markers (Fig. 7i, right) were detected in PNPT1-knockout mice. Given that PNPT1 deficiency likely disrupts mt-dsRNA homeostasis and mitochondrial function, we examined the mitochondrial structure in

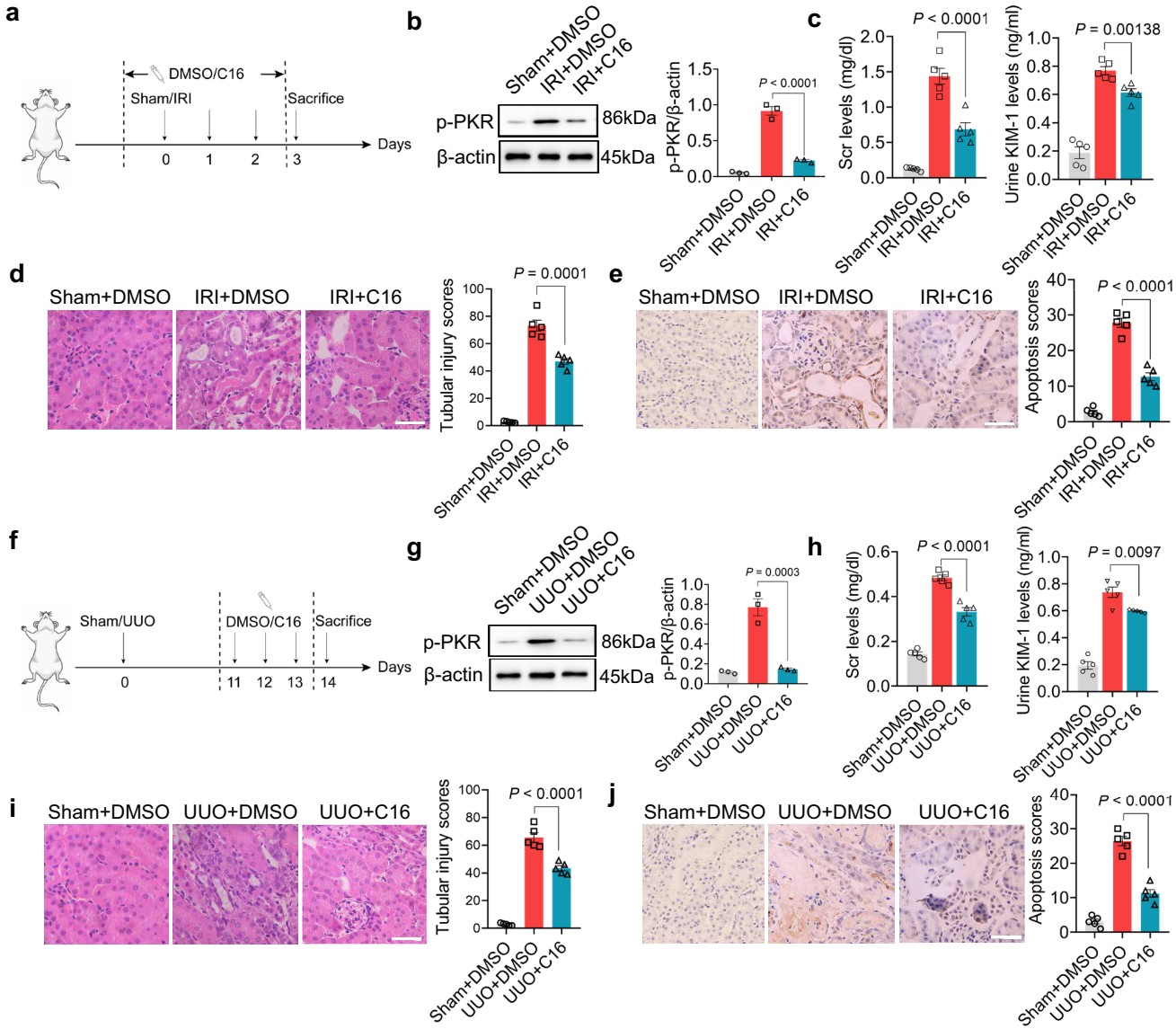

**Fig. 6 | Inhibition of PKR protected renal tubule against injury in experimental mouse models. a** Schematic of experimental approach in IRI mouse model with or without C16 treatment. **b** Left: WB of renal tubular p-PKR levels in IRI mouse model with or without C16 treatment. Right: quantification of renal tubular p-PKR levels (3 mice/group). **c** Left: Scr in IRI mouse model with or without C16 treatment. Right: urinary KIM-1 levels in IRI mouse model with or without C16 treatment (5 mice/group). **d** Left: H&E staining of kidney tissue sections from IRI mouse model with or without C16 treatment. Right: quantification of kidney disease score (5 mice/group). **e** Left: TUNEL assay of kidney tissue sections from IRI mouse model with or without C16 treatment. Right: quantification of apoptosis in kidney tissues (5 mice/group). **f** Schematic of experimental approach in UUO mice with or without C16 treatment. **g** Left: WB of renal tubular p-PKR levels in in UUO mice with or without

C16 treatment. Right: quantification of renal tubular p-PKR levels (3 mice/group). **h** Left: Scr in UUO mouse model with or without C16 treatment. Right: urinary KIM-1 levels in UUO mice with or without C16 treatment (5 mice/group). **i** Left: H&E staining of kidney tissue sections from UUO mice with or without C16 treatment. Right: quantification of kidney disease score (5 mice/group). **j** Left: TUNEL assay of kidney tissue sections from UUO mice with or without C16 treatment. Right: quantification of apoptosis in kidney tissues (5 mice/group). Scale bars, 50 μm. The above experiments were successfully repeated three times. One-way ANOVA with Tukey's multiple comparisons test was performed in **b–e**, **g–j** and the results were presented as mean ± SEM. Images of mouse in **a**, **f** were created with BioRender.com. Source data are provided as a Source Data file.

renal tubular cells from PNPT1-knockout and WT mice using transmission electronic microscope (TEM). As shown in Fig. 7j, PNPT1-knockout mice displayed more significant matrix destruction, cristae fragmentation, cavity enlargement and mitochondrial swelling compared to WT mice. Masson staining of kidney tissue sections from WT and PNPT1-knockout mice also showed a significantly higher degree of renal fibrosis in PNPT1-knockout mice than in WT mice (Fig. 7k). As renal tubular injury eventually leads to cell apoptosis, we assessed the level of cell apoptosis in the mouse kidney. As shown by the TUNEL

assay, PNPT1-knockout mice exhibited significantly higher apoptosis levels in renal tubular cells than in WT mice (Fig. 7l).

## Discussion

This study reveals that reduction of renal tubular PNPT1 under various renal dysfunctions, including ATN, DN, LN, IgAN, FSGS, etc., is a key causative factor of renal tubular injury. As depicted in Fig. 4f, stimulated by HG, TGFβ or LPS, all three factors of which are tightly related to renal tubular cell injury[38,39], renal tubular cell PNPT1 is drastically

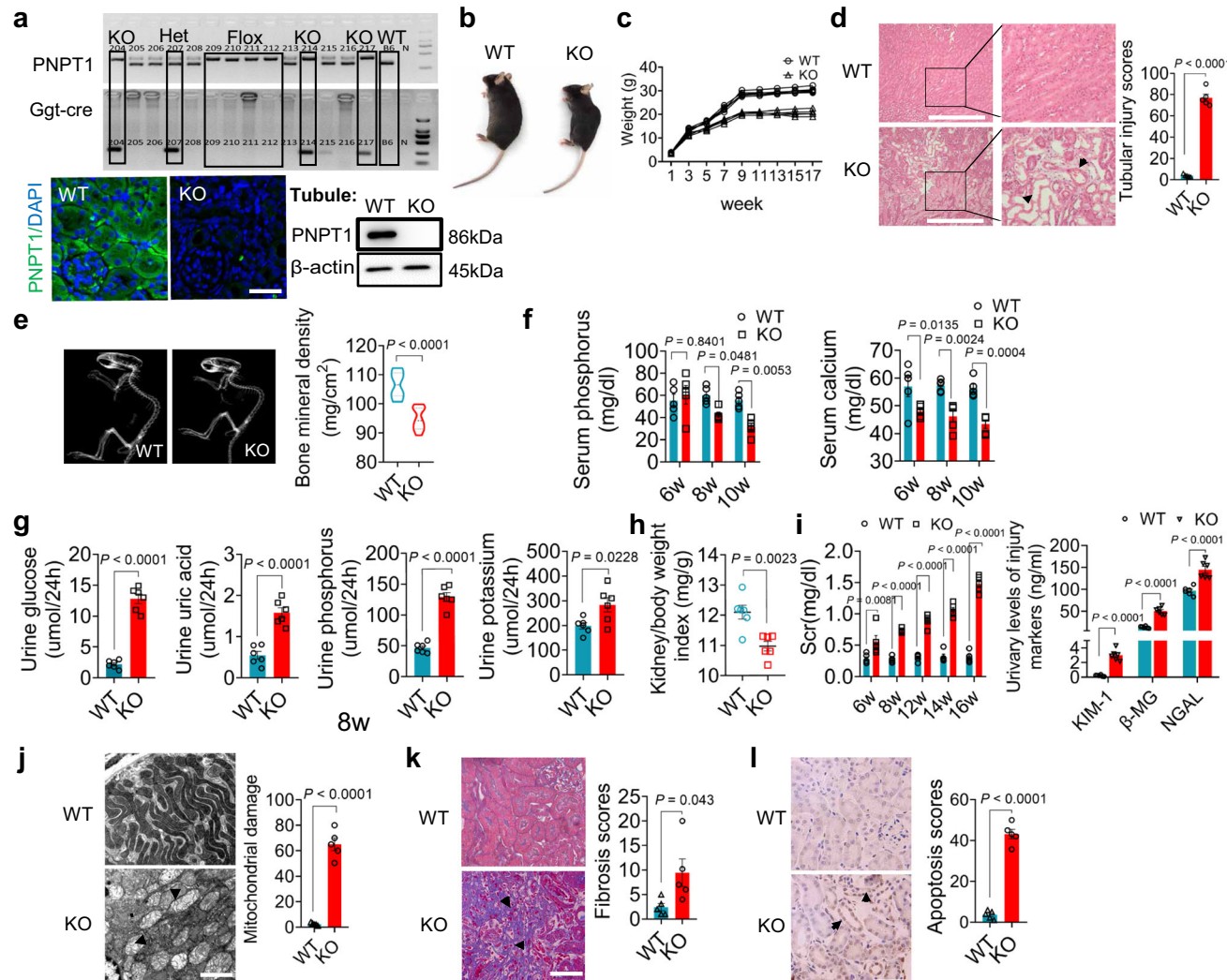

**Fig. 7 | Renal tubular-specific PNPT1[-/-] mice displayed impaired reabsorption and renal tubular injury. a** Top: genotype identification and PNPT1 levels in WT and renal tubular-specific PNPT1-KO (KO) mice. Bottom: immunofluorescence labeling and WB analysis of PNPT1 in WT and KO renal tubules. **b** Image of WT and KO mice. **c** Weight monitoring of WT and KO mice (7 mice /group). **d** Tubular injury score in WT and KO mice (5 mice/group, 8w). Arrows: injured tubules. **e** Left: Bone mineral density scanning of WT and KO mice. Right: quantification of bone mineral density (10 mice/group, 8w). **f** Levels of serum calcium and phosphorus (5 mice/group, 6-10w). **g** Urine level of glucose, uric acid, phosphorus and potassium in WT and KO mice (6 mice/group, 8w). **h** Kidney/body weight index (6 mice/group, 8w). **i** Serum creatinine (Scr, left) and levels of renal tubular injury markers (right) in WT and KO mice (5 mice/group, 8w). **j** Left: TEM image of mitochondrial damage in

renal tubule from WT and KO mice. Right: quantification of renal tubular mito-chondrial damage (5 mice/group, 8w). Arrows: injured mitochondria. **k** Left: Masson staining in kidney tissue sections from WT and KO mice. Right: quantification of renal fibrosis (5 mice/group, 8w). Arrows: fibrotic tubules. **l** Left: TUNEL assay of kidney tissue sections from WT and KO mice. Right: quantification of renal tubular apoptosis (5 mice/group, 8w). Arrows: apoptotic tubules. Scale bars in **a, d, k,** and **l,** 50 μm. Scale bar in **j,** 2 μm. The above experiments were successfully repeated three times. Two-tailed unpaired *t* test was performed for the statistical analyses in (**d, e, g, h, j–l**), Two-way ANOVA with Sidak's multiple comparisons test was performed in **f, i,** and the results were presented as the mean ± SEM. Source data are provided as a Source Data file.

reduced. The PNPT1 reduction will lose control of mitochondrial RNA homeostasis, resulting in leakage of mt-dsRNAs into the cytoplasm where they activate PKR and initiate the PKR-eIF2α signaling axis to terminate general protein translation. Suppression of protein synthesis in renal tubular cells then leads to cell injury and eventually apoptosis.

Given its primary function of reabsorption which requires a great amount of energy produced by mitochondria, renal tubular cells indeed consist of a much higher number of mitochondria than other renal cells, such as podocytes[10]. No surprising, renal tubular cells, particularly proximal tubule cells (PTCs), are very vulnerable to mitochondrial impairment. It has been widely reported that various mitochondrial dysfunctions are tightly linked to various renal tubular cell diseases[11,12]. However, most studies that investigated the role of mitochondria in maintaining renal tubular cell function have been focused on metabolic

reprograming, as well as oxidative stress factors such as reactive oxygen species (ROS)[48]. Our studies here demonstrate that the homeostasis and dynamic turnover of mt-dsRNAs are also key factors for the fate of renal tubular cells. As an essential threat to most mammalian cells, dsRNAs can be recognized by various RNA sensors, including RIG-I/MDA5, which triggers a type I interferon response[31], and PKR, which initiates an integrated stress response (ISR) that terminates general protein synthesis. As cytosolic mt-dsRNAs induced by silencing renal tubular cell PNPT1 rapidly suppressed general protein synthesis (Fig. 4a), our results indicate that PKR-eIF2α-mediated ISR plays a major role in mt-dsRNA-induced renal tubular injures. In fact, as a conservative anti-viral host defense mechanism, the ISR induced by viral dsRNAs has been exten-sively characterized[49]. However, recent studies demonstrate that ISR also modulates mammalian cell proliferation and survival because it can be initiated by endogenous dsRNAs, particularly mt-dsRNAs[18]. In

mammals, mitochondrial DNA can generate large amounts of overlapping transcripts, which are capable of forming long dsRNAs. By detecting cytosolic mt-dsRNAs or nuclear dsRNAs, Kim et al. showed that PKR served as a sensor for mitochondrial and nuclear signaling cues in regulating cellular metabolism[18]. Our results demonstrate that cytosolic mt-dsRNAs in injured renal tubular cells indeed bind to and activate PKR, which in turn, phosphorylates eIF2α, leading to the termination of protein translation.

Supporting the role of ISR induced by mt-dsRNA-PKR-eIF2α axis in renal tubular injury, our intervention studies using both IRI and UUO mouse renal tubular injury models show that compensating the PNPT1 reduction in mouse renal tubules can markedly decrease cytosolic mt-dsRNA levels (Fig. 5b, f) and prevent renal tubular injures, as evidenced by reduced Scr, urinary KIM-1 levels (Fig. 5c, g) and injury index (Fig. 5d, h). Moreover, directly inhibiting renal tubular cell PKR activity by C16 also strongly protects mice against renal tubular injures induced by IRI and UUO procedures (Fig. 6). The protective role of PKR inhibition is in agreement with a previous report that PKR deficiency decreases congestive heart failure induced by systolic overload[50].

As an evolutionarily conservative 3′-to-5′ exoribonuclease that is involved in RNA processing and degradation, mitochondria-associated PNPT1 controls the homeostasis of mitochondria RNAs[51]. Supporting this, reduction of PNPT1 in renal tubular cells, results in elevation of cytosolic mt-dsRNAs (Fig. 3d). The defect of PNPT1 has been shown to result in mitochondrial dysfunction and various neurologic problems such as autosomal recessive deafness[52,53]. Although no such PNPT1 deficiency or mutation has been linked to renal diseases, our results clearly show that PNPT1 activity is required to maintain the normal function of renal tubular cells. A significant reduction of renal tubular PNPT1 is found in different renal diseases which are associated with various degrees of renal injury (Fig. 2). Although the underlying molecular basis remains unknown, mechanistic studies indicate that PNPT1 reduction in renal tubular cells can be achieved by renal tubular cell injury factors TGFβ1, hyperglycemia and LPS (Fig. 3a). Finally, the essential role of PNPT1 in maintaining normal renal tubular function under various stress conditions is validated in renal tubular-specific PNPT1-knockout mice. Compared to WT mice, PNPT1-knockout mice displayed significant renal tubular injury (Fig. 7d), accompanied with higher levels of Scr and renal tubular injury markers (Fig. 7i). As calcium filtered by the kidney is efficiently (>98%) reabsorbed by renal tubules to maintain the serum concentration[54], damage of renal tubular cells, particularly PTCs, would result in impairment of calcium reabsorption and significant bone loss. Indeed, we found that PNPT1-knockout mice displayed severe impaired reabsorption, as evidenced by a lower body weight (Fig. 7c), a severe bone structure deformation, a markedly lower bone mineral density (Fig. 7e) and significant loss of serum calcium and phosphorus (Fig. 7f). Direct analysis of renal tubular cell mitochondria at ultrastructural level confirms that PNPT1 deficiency markedly disrupts mt-dsRNA homeostasis and mitochondria function, causing significant matrix destruction, cristae fragmentation, cavity enlargement and mitochondrial swelling (Fig. 7j).

In conclusion, these findings reveal that reduction of renal tubular PNPT1 under various injury conditions is a major cause of renal tubular injury, and provide PNPT1 as a therapeutic target for protecting against renal tubule injury.

## Methods

### Statement
All protocols concerning the use of patient samples in this study were approved by the Human Subjects Committee of Jinling Hospital, Nanjing University School of Medicine (2019NZKYKS-008-01). A written informed consent was obtained from each patient whose sample is used in the study. Animal study was approved by Animal Ethical and Welfare Committee of Nanjing University (IACUC-2112001). Mice were housed, bred and studied in accordance with approved protocols.

Mouse euthanasia was done first by anaesthesia and then followed by cervical dislocation.

### Cells, antibodies, and reagents
HK2 cells were obtained from American Tissue and Cell Center (ATCC, CRL-2190) and grown as a monolayer in DMEM/F12 (Gibco) supplemented with 10% FBS (Gibco), 100 U/ml penicillin (Gibco), 100 μg/ml streptomycin (Gibco) and ITS (Sigma-Aldrich). Primary antibodies for dsRNA mAb J2 (Scions, 10010500, 1:200), PNPT1 (Abcam, ab96176, 1:200/2000), AQP1 (Abcam, 168387, 1:200), synaptopodin (Abcam, ab224491, 1:100), podocin (Sigma-Aldrich, P0372, 1:100), PKR (Abcam, ab32506, 1:1000), p-PKR (Abcam, ab32036, 1:1000), eIF2α (CST, 5324 S, 1:1000), p-eIF2α (CST, 3398 S, 1:1000), ATF4 (Abcam, ab184909), COX IV (Abcam, ab14744, 1:1000), LaminB1 (Abcam, ab65986, 1:1000), and α-tubulin (Proteintech, 11224-1-AP, 1:2000) were used for immunofluorescence and western blot analysis; Secondary antibodies including goat anti-mouse Alexa Fluor 488 (Invitrogen, A11001, 1:1000), donkey anti-rabbit Alexa Fluor 594 (Invitrogen, A21207, 1:1000), donkey anti-rabbit Alexa Fluor 488 (Invitrogen, A21206, 1:1000), goat anti-mouse Alexa Fluor 594 (Invitrogen, A11005, 1:1000), goat anti-mouse IgG-HRP (Santa Cruz Biotechnology, sc-2005, 1:1000) and goat anti-rabbit IgG-HRP (Santa Cruz Biotechnology, sc-2004, 1:1000) were used correspondingly. DAPI was purchased from Santa Cruz Biotechnology (sc-3598, 1:1000). TGFβ (R&D Systems, 7754), LPS (Sigma-Aldrich, L2630), HG (Sigma-Aldrich, G7021) were used. Digitonin (Sigma-Aldrich, D141) were used for membrane permeabilization. KIM-1 (Abcam, ab213477), NGAL (Abcam, ab118901), β2-MG (Abcam, ab223590) ELISA quantitation kits, creatinine kit (Sigma-Aldrich, MAK080), calcium assay kit (Sigma-Aldrich, MAK022) and phosphorus microplate assay kit (Absin, 580107) were purchased for renal tubular injury assessment. Lipofectamine RNAi MAX (Invitrogen, 13778150) were used for transfection. PKR inhibitor (C16) was obtained from Sigma-Aldrich (I9785) and dissolved in DMSO prior to experiment. IMT1 was obtained from MCE (HY-134539) and dissolved in DMSO prior to experiment. FITC-Annexin V Apoptosis Detection Kit (Invitrogen, V13241) and TUNEL Bright-Green apoptosis detection kit (Vazyme, A112) were used for apoptosis analysis. Collagenase D (1 mg/mL, C6885), protease (1 mg/mL, P6911) and DNaseI (1 U/mL, D5025) were purchased from Sigma-Aldrich. Click-iT® Plus OPP Protein Synthesis Assay Kits were purchased from Life Technologies (C10456). PierceTM Classic IP Kit were purchased from Thermo Fisher Scientific (26146). Ribo™ Fluorescent In Situ Hybridization Kit were purchased from Ribo (C10910). PNPT1 AAV were purchased from OBiO (H131559). siPNPT1 plasmid (siG000087178A-1-5) and siPKR plasmid (siB161220105622-1-5) were purchased from Ribo.

### Kidney biopsy
Kidney tissues were obtained through percutaneous renal biopsy from patients with ATN (n = 12, 5 males and 7 females), DN (n = 14, 8 males and 6 females), IgAN (n = 16, 9 males and 7 females), LN (n = 15, 5 males and 10 females), MN (n = 13, 8 males and 5 females) and FSGS (n = 15, 6 males and 9 females), as well as 5 non-renal tubular injury controls (3 males and 2 females). Non-tubular injury paracancerous renal tissues (n = 5) were obtained from renal clear cell cancer patients. All patients were diagnosed based on renal biopsies at the National Clinical Research Center of Kidney Diseases, Jinling Hospital, Nanjing University School of Medicine.

### Mouse strains and experimental renal tubular injury models
Male C57BL/6 J (8 weeks, 22–25 g) and renal tubular cell-specific PNPT1-knockout (KO) mice (1-18 weeks, male) were obtained from the Model Animal Research Center of Nanjing University. In brief, PNPT1-flox homozygous were first generated using the CRISPR/Cas9 system, and PNPT1-KO mice were obtained by mating

PNPT1-flox homozygous and Ggt-Cre mice. All mice were back-crossed to a C57BL/6 J background. All mice were fed with lab diet (SWC9101) under standard conditions of constant temperature ($22 \pm 1\,°C$), humidity (relative, 30%), in a pathogen-free facility and exposed to a 12-h light/dark cycle. Inductions of kidney IRI and UUO were performed as described previously[55]. Briefly, after mice were anesthetized, a midline abdominal incision was made and bilateral renal pedicles were clipped for 35 min using microaneurysm clamps. After removal of the clamps, reperfusion of the kidneys was visually confirmed. The incision was then closed and the animal was allowed to recover. During the ischemic period, body temperature was maintained between $37\,°C$ using a temperature-controlled heating system. Blood urine and tissue samples were obtained at 1d and 3d post-IRI. In the UUO model, the left ureter was ligated at two points and cut between them. Mice were sacrificed 7d and 14d after the procedure. To overexpress PNPT1 in mouse renal tubule, PNPT1-expressing AAV (PNPT1 AAV) and control AAV (Ctrl AAV) were generated (OBIO Technology). $1 \times 10^{13}$ VG/ml PNPT1 AAV or Ctrl AAV were in situ injected into mouse kidney (20 µl per mouse) during IRI model establishment. To block PKR activity, PKR inhibitor (C16, 500 µg/ml) was injected intraperitoneally three days in a row following IRI or UUO procedure. For monitoring mouse renal tubular injury, urine samples were collected over 24 h once a week. Renal tubular injury markers including urinary creatinine, KIM-1, NGAL and β2-MG were measured. To suppress mtRNA expression, IMT1 (30 mg/kg) was injected intraperitoneally three days in a row after IRI or UUO procedure. The kidney tissues were collected for H&E staining and TUNEL analysis. To determine tubular injury, defined as loss of the brush border, compensatory tubular dilatation and detachment, tubular apoptosis and cellular casts, a semiquantitative scoring method was used. Score 0 represents injury area less than 10%, whereas 1, 2, 3 and 4 represent the injury involving 10–25, 25–50, 50–75 and >75% of the renal tubular area, respectively. At least 12 randomly chosen fields under the microscope were evaluated for each mouse tissue section and an average score was calculated. To monitor renal tubular reabsorption, bone mineral density (BMD) was evaluated by Ultra Focus DXA (Faxitron), while urinary calcium and urinary phosphorus were detected with the calcium assay and phosphorus microplate assay kits, respectively.

## Cell treatment
For mimicking renal tubular cell injury in AKI and CKD conditions, HK2 cells were incubated with 10 ng/mL TGFβ, 75 µg/mL LPS or 40 mM HG for various indicated times. For cell transfection, HK2 cells were seeded in a 6-well plate and cultured to a confluence of 70–80%. Cells were then transfected with PNPT1 siRNAs (final concentration of 20 nM) or control oligonucleotide with Lipofectamine RNAi MAX according to the manufacturer's instructions. For blocking PKR-eIF2α pathway in siPNPT1 HK2 cells, p-PKR inhibitor C16 (0.5 µM, 4 h) dissolved in 0.1% DMSO was used. To knockdown PKR expression, siRNAs was transfected using Lipofectamine RNAi MAX following the manufacturer's instructions. To suppress mtRNA synthesis, HK2 cells were treated with 0.1 mM of IMT1 for 72 h.

## Apoptosis assay
Cell apoptosis was determined using an FITC-Annexin V Apoptosis Detection Kit. Briefly, $2 \times 10^5$ cells were resuspended in 0.5 ml binding buffer and incubated with FITC-Annexin V and PI for 15 min in the dark. Data were acquired with a Attune NxT (Thermo Fisher Scientific) flow cytometer and analyzed by FlowJo, and apoptotic cells were identified from the Annexin V-positive, PI-negative fractions. Cell apoptosis in paraffin-embedded kidney sections (4µm thickness) was detected by the TUNEL assay. Tissue sections were subjected to dewaxing and hydration and then to apoptosis analysis using the TUNEL Bright-Green apoptosis detection kit. Quantification of TUNEL staining

was determined by counting the percentage of TUNEL-positive tubular cells.

## Histological and immunofluorescence analysis
H&E staining of 4-µm-thick formalin-fixed, paraffin embedded kidney tissue sections was performed according to standard protocols. Images were captured by microscopy (Olympus BX53). In each kidney, the numbers of damaged tubules were counted separately in 10 non-overlapping fields observed at ×400 magnification. Renal tubular injury scores expressed as the percentage of damaged tubules, and the mean values were calculated (GraphPad Prism). For immunofluorescence staining of kidney tissue sections, heat-induced antigen retrieval was performed by heating sections in 1 mM EDTA at $95\,°C$ in a pressure cooker for 20 min and then 20 min of cooling at room temperature. Sections were permeabilized with 0.1% Triton X-100 in 1× PBS. Sections were incubated for 30 min at room temperature with blocking buffer 5% BSA in 1× PBS and subsequently incubated with primary antibodies including anti-dsRNA mAb J2, anti-PNPT1, anti-AQP1, and anti-podocin antibodies for 3 h. Sections were washed in 1× PBS three times and then incubated with secondary antibodies including goat anti-mouse Alexa Fluor 488, donkey anti-rabbit Alexa Fluor594, donkey anti-rabbit Alexa Fluor 488, and goat anti-mouse Alexa Fluor 594 antibodies for 1 h at room temperature followed by 3 washes with 1× PBS. The sections were then mounted in Prolong Diamond Antifade Mountant with DAPI. Confocal images were taken using a confocal microscope (Zeiss LSM 880) with ZEN 3.1 blue edition software. Fluorescence intensities were quantified using ImageJ 1.44p software.

## RNA FISH
To visualize heavy and light strands of *mt-ND5* RNA, RNA fluorescent in situ hybridization (RNA-FISH) was performed using Ribo™ Fluorescent In Situ Hybridization Kit. The pretreatment, hybridization, and signal amplification were done following the manufacturer's instructions. Zeiss LSM 880 confocal microscope was used for visualization. Fluorescence intensities were quantified using ImageJ 1.44p software.

## Transmission electronic microscopy (TEM)
Renal cortices were collected and dissected into 1-mm³ pieces and fixed in 2.5% glutaraldehyde, followed by post-fixation in 2% osmium tetroxide, dehydrated in graded series of acetone and ethanol, and embedded in epoxy resin. Ultrathin sections (80–90 nm) were stained for 15 min in 5% uranyl acetate, followed by 0.1% lead citrate for 5 min. Electron micrographs were obtained and analyzed using a Hitachi 7500 transmission electron microscope.

## Western blot analysis
Mouse renal tubules were isolated. Briefly, the renal cortex was cut into small cubes of ~1 mm³ and digested with collagenase D, protease and DNaseI in HBSS for 15 min at $37\,°C$. The digested tissue was screened through a 100 µm filter. Glomeruli-containing Dynabeads (diameter 4.5 µm) were adsorbed using a magnetic particle concentrator and tubulars were gathered from the supernatant after adsorption. Protein extracts from renal tubules or HK2 cells were resolved by SDS-PAGE before being transferred onto the polyvinylidene fluoride (PDVF) membranes. The membranes were blocked and incubated with primary antibodies against PNPT1, PKR, p-PKR, eIF2α, p-eIF2α, ATF4, COX IV, LaminB1, and α-tubulin followed by 6 washes and incubation with secondary antibodies goat anti-mouse IgG-HRP or goat anti-rabbit IgG-HRP. Protein band intensities were quantified using ImageJ 1.44p software. Uncropped scans of all gels are provided in Source data file with this paper.

## Protein synthesis assay
Whole protein synthesis of HK2 cells treated with or without siPNPT1 at 12 and 24 h post-transfection was assayed by Click-iT® Plus OPP Protein

Synthesis Assay Kits. Images were taken using a 2-photon laser confocal microscope (Zeiss LSM 880). Mean intensities were quantified using ImageJ 1.44p software.

## Detection of cytosolic mt-dsRNAs

HK2 cells ($5 \times 10^8$) were resuspended in 500 μl buffer containing 150 mM NaCl, 50 mM HEPES, 25 μg/ml digitonin, pH7.4. The homogenates were incubated end-over-end for 10 min to allow selective plasma membrane permeabilization, then centrifuged at $1000\,g$ for 5 min to pellet intact cells. The supernatants were transferred to fresh tubes and spun at $17,000\,g$ for 10 min to pellet any cellular debris, yielding cytosolic preps free of nuclear, mitochondrial and ER contamination. The dsRNAs in cytosolic fraction were immunoprecipitated by J2 antibody followed by PierceTM Classic IP Kit. The dsRNA was then extracted and subjected to strand-specific RT-qPCR analysis. Reverse transcription primers and RT-qPCR primers were designed to target the specific genes (*mt-ND4-6*, *mt-CO1* or *mt-CYB*) (Supplementary Table. 1). The mt-dsRNA levels were normalized to β-actin mRNA level.

## Statistical analysis

Data were presented as mean ± SEM. Statistical analysis was performed in GraphPad Prism 8.0. All data were obtained from at least 3 independent experiments. Details of the statistical tests were described in the individual figure legends. Data with *P* value <0.05 were considered statistically significant. Sample size base on the data in our previous study and a given power (>0.8) using G-power software 3.1.

## Reporting summary

Further information on research design is available in the Nature Portfolio Reporting Summary linked to this article.

# Data availability

All relevant data generated for this study were included in the article/Supplementary Material. All the data analyzed during this study are available from the corresponding authors upon request. Source data are provided with this paper.

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

## Acknowledgements
This work was supported in part by grants from National Key R&D Program of China (2018YFA0507100, K.Z.) and National Natural Science Foundation of China (82170692, K.Z.). Kidney samples were obtained from Renal Biobank of National Clinical Research Center of Kidney Diseases, Jiangsu Biobank of Clinical Resources, a part of The Open Project of Jiangsu Biobank of Clinical Resources (JSRB2021-03).

## Author contributions
K.Z., Z.L. and L.L. conceived and designed the experiments; Y.Z., M.Z., W.W., S.Q., M.L., W.Y., H.L. and W.R. performed experiments; Y.Z. and K.Z. analyzed data; Z.L., C.Z., M.Z. and X.Z. contributed technical or material support; K.Z. and Y.Z. wrote the paper.

## Competing interests
The authors declared no competing interests.
