## [Peer Review File · Nature Communications]

Polynucleotide phosphorylase protects against renal tubular injury via blocking mt-dsRNA-PKR-eIF2 α axisREVIEWER COMMENTS

Reviewer #1 (Remarks to the Author):

The manuscript by Zhu et al. investigated the role of mt-dsRNAs in renal tubular atrophy. The authors found the reduced expression of PNPT1, an essential component of mt-dsRNA degradosome, in both patients and mice tissue samples. The decreased expression of PNPT1 subsequently resulted in the cytosolic release of mt-dsRNAs, which resulted in global translational suppression and apoptosis through PKR activation. Importantly, the authors showed that the overexpression of PNPT1 or inhibition of PKR resulted in decreased renal tubular injuries.

Overall, the manuscript is well written and the analysis is sound. However, this reviewer has a number of concerns regarding the experimental evidence and claims by the authors. Specifically,

Major comments:

1) Recently, numerous studies reported the involvement of mt-dsRNAs in various human diseases such as alcohol-induced liver damage (Lee et al., *Hepatology* 2020), Huntington's disease (Lee et al., *Neuron* 2020), and osteoarthritis (Kim et al., *Cell Reports* 2022). Please cite these studies.

2) J2 staining can detect cellular dsRNAs other than mt-dsRNAs. In fact, a study by Kim et al. (*Mol Cell*, 2018) showed that most of the RNAs recognized by the J2 antibody were Alu RNAs. Therefore, in Figure 1, the authors must confirm that the increased J2 staining is due to mt-dsRNAs via RNA-FISH and/or qPCR.

3) In Figure 2b, how many images and how many patients were used? Please specify

4) In Figure 3d, please indicate the normalization control. In addition, the increased mtRNA could be due to decreased expression of the housekeeping gene that was used as a normalization control. Therefore, the authors must confirm the increased mt-dsRNA expression via RNA-FISH.

5) Figure 3d finally shows qPCR results of cytosolic mtRNA expression. However, this data does not reflect mt-dsRNAs. The authors must perform strand-specific reverse transcription and show the increased expression of both heavy and light-strand mtRNAs to infer a potential increase in mt-dsRNA expression.

6) What happens to protein synthesis when PKR is inhibited by C16 in PNPT1-deficient cells?

7) Please show that overexpression of PNPT1 (PNPT1-AAV) successfully resulted in mitochondrial expression of PNPT1 protein.

8) C16 can also inhibit other kinases including CDKs. Please confirm that knockdown of PKR results in the same phenotypes as those of C16.

9) Although the authors showed that inhibition of PKR using C16 rescued renal tubular injuries, most direct evidence will be suppressing mtRNA expression directly using IMT1 (Bonekamp et al., *Nature* 2020). Please analyze the kidney tissues of IRI and UUO mice after administering them with IMT1.

Minor comments:

1) Line 69: Please remove "as".

2) Line 69-70: Most endogenous dsRNAs are derived from Alu elements and thus are not predominantly confined to the mitochondria.

3) Line 122: Please remove "degree".

4) Line 126: Again, endogenous dsRNAs are not predominantly derived from mitochondria.

5) Line 145: A statement that PNPT1 expression can be modulated by IFN needs a reference.

Reviewer #2 (Remarks to the Author):

The manuscript by Zhu et al addresses the hypothesis that the effects of reduced PNPT1-induced mitochondrial dsRNA release to the cytosol, enhances tubular injury in acute, and perhaps chronic kidney disease pathophysiology. While the studies are exhaustive, there are problems with the methods and interpretation of the data, as described below.

Major criticisms

1. A major goal is to characterize pathophysiologic processes that regulate tubular atrophy, which is generally considered to be inexorable and accompanied by interstitial fibrosis, as part of chronic kidney disease progression. However, most of the data are in acute (and potentially reversible) models (especially ATN from IRI), which are not usually characterized by tubular atrophy. In the Figure 1c images, with the possible exception of DN, the interstitium does not appear to be increased (though a counterstain for scar matrix would be helpful), and tubules appear intact, suggesting absence of tubular atrophy. It might make more sense to frame the project in terms of acute tubular injury, and delete the association with tubular atrophy or chronic kidney disease.
2. The UUO model would be expected to cause injury predominately to collecting ducts and distal tubules, whereas AQP1 labels proximal tubules. A related issue is that mice with Cre under the control of a GGT promoter, crossed with floxed mice, would result in proximal tubule-specific knockout mice, i.e., not all tubules. Some explanation is required regarding how UUO effects would be reversed in these mice with injury to non-proximal tubules. A separate issue with the UUO model is that it is customary to ligate the ureter for a finite time period, then release the ligation, and examine residual effects. In the Methods (lines 377-378) the ureter is ligated and cut, which results in irreversible obstruction. It is difficult to envision how any intervention would rescue the phenotype of a permanently obstructed kidney.
3. The use of PI in the apoptosis assays confounds the interpretation for several reasons. First, it is generally used as a marker for necrosis, or perhaps late apoptosis in PI-positive + annexin V-positive cells. More importantly, PI can yield false-positive readings due to labeling of cytosolic RNA (PMID: 20381494). A quick solution would be to merely count the annexin V-positive, PI-negative cells. Alternatively, it may be necessary to pre-treat cells with RNAses or choose a completely different apoptosis assay.
4. Fanconi syndrome is defined by enhanced urinary excretion of glucose, phosphorus, bicarbonate, uric acid, potassium and amino acids. These analytes should be measured to justify the claim that PNPT1 deletion phenocopies Fanconi syndrome. Although mild hypocalcemia may be observed with Fanconi syndrome, osteopenia is generally associated with hypophosphatemia. If proximal tubule deletion of PNPT1 does indeed cause Fanconi syndrome, that is an interesting finding. However, this is not a feature of IRI-induced ATN or urinary tract obstruction. To exploit the PNPT1 knockout mice in a manner that is relevant to ATN and UUO, it might be more useful to characterize the PTPN1 heterozygotes.
5. There are several key assays, for which the criteria/scoring system were not defined. The first is tubular injury (Figures 1 and 6). The associated box plots corresponding to tubular atrophy in Figures 1d and 1f were also not defined. For Figure 1 and 5 dsRNA immunohistochemical quantification, how was the threshold established for positive vs. negative? And is the denominator for relative staining all tubules, tubular epithelial cells, proximal tubule cells?

Minor criticisms

1. Modest editing is required for English.
2. Suspect the glomerulus and tubule labels for Extended data 2b western blots are reversed.
3. The number of samples used to generate Figure 2b is not stated.
4. For detection of cytosolic mt-dsRNAs, verification that lysates are free of mitochondrial contamination should be shown.
5. Although Figures 5a and 5e demonstrated PNPT1 AAV-injected mice do not have decreased PNPT1 expression after IRI or UUO, suggesting that the injection resulted in PNPT1 overexpression, this should be verified with western blot lanes corresponding to Sham + AAV PNPT1.

6. Figure 7a genotyping figure requires more explanation. In particular, which lanes are WT and which are KO, and what is the significance of the boxed lanes? What are the expected band patterns for each genotype? In addition, please state whether there is any phenotype in the GGT-Cre only and floxed only mice.

Jeffrey R. Schelling

Point-to-point response letter

Reviewer 1#

1. Recently, numerous studies reported the involvement of mt-dsRNAs in various human diseases such as alcohol-induced liver damage (Lee et al., Hepatology 2020), Huntington's disease (Lee et al., Neuron 2020), and osteoarthritis (Kim et al., Cell Reports 2022). Please cite these studies.

Response: We thank reviewer for pointing out this. Accordingly, we have cited these studies in the Introduction of our revision (Page 5, marked in red). It reads as: 'Recently, numerous studies reported the involvement of mt-dsRNAs efflux into cytoplasm to activate the innate immune system in various human diseases, including alcohol-induced liver damage¹⁵, Huntington's disease¹⁶, and osteoarthritis¹⁷.

2. J2 staining can detect cellular dsRNAs other than mt-dsRNAs. In fact, a study by Kim et al. (Mol Cell, 2018) showed that most of the RNAs recognized by the J2 antibody were Alu RNAs. Therefore, in Figure 1, the authors must confirm that the increased J2 staining is due to mt-dsRNAs via RNA-FISH and/or qPCR.

Response: We greatly appreciate reviewer's constructive suggestion. Indeed, J2 antibody has affinity to all double stranded RNAs. According to reviewer's comment, we performed the strand-specific reverse transcription (qPCR) (new Fig. 1, g and h) of mitochondria double stranded RNAs, mt-ND4, mt-ND5, mt-ND6, mt-CO1 and mt-CYB. The results showed that both heavy and light transcripts of these mt-dsRNAs in the cytosol were markedly elevated in IRI mice tubules. We also performed RNA-FISH of mt-ND5 (light and heavy chain) in HK2 cells with or without PNPT1 reduction. As shown in new Fig. 3f and 3g, PNPT1 reduction caused significant accumulation of mt-ND5 light and heavy chain in cytoplasm of HK2 cells. Our results clearly show the mt-dsRNA efflux into cytoplasm under tubular injury condition.

3. In Figure 2b, how many images and how many patients were used? Please specify.

Response: ATN (n=12, 5 males and 7 females), DN (n=14, 8 males and 6 females), IgAN (n=16, 9 males and 7 females), LN (n=15, 5 males and 10 females), MN (n=13, 8 males and 5 females) and FSGS (n=15, 6 males and 9 females), as well as 5 non-renal tubular injury controls (3 males and 2 females). Images of 10 randomly selected regions per sample were obtained and analyzed.

4. In Figure 3d, please indicate the normalization control. In addition, the increased mtRNA could be due to decreased expression of the housekeeping gene that was used as a normalization control. Therefore, the authors must confirm the increased mt-dsRNA expression via RNA-FISH.

Response: Levels of cytosolic mtRNAs are normalized against β -actin mRNA level. According to reviewer's comment, we performed fluorescent RNA-FISH experiment to confirm the increase of mt-dsRNA expression in cytoplasm. As shown in new Fig. 3f and 3g, a significantly increased cytosolic expression of both mt-ND5 sense (ND5 heavy) and antisense (ND5 light) transcripts were observed in HK2 cells with PNPT1 reduction by siPNPT1.

5. Figure 3d finally shows qPCR results of cytosolic mtRNA expression. However, this data does not reflect mt-

dsRNAs. The authors must perform strand-specific reverse transcription and show the increased expression of both heavy and light-strand mtRNAs to infer a potential increase in mt-dsRNA expression.

Response: To show that the cytosolic dsRNAs were indeed mt-dsRNAs, we performed the strand-specific reverse transcription of mt-ND4, mt-ND5, mt-ND6, mt-CO1 and mt-CYB. The results showed that both heavy and light transcripts of these mt-dsRNAs in the cytosol of renal tubules were elevated in mice with IRI compared to those in control mice (Fig. 1, g and h). We also performed RNA-FISH of mt-ND5 (light and heavy chain) in HK2 cells with or without PNPT1 reduction, and results showed that PNPT1 reduction caused significant accumulation of cytosolic mt-ND5 light and heavy chain in HK2 cells (new Fig. 3f and 3g). Our results confirm the mt-dsRNA efflux into renal tubular cytoplasm under tubular injury condition.

6. What happens to protein synthesis when PKR is inhibited by C16 in PNPT1-deficient cells?

Response: To answer reviewer's question, we monitored the protein synthesis in PNPT1-deficient cells treated with or without C16. As shown in our new Extended Data, figure 6a, protein synthesis was rescued in PNPT1-deficient cells when cells were treated with C16. The result confirms that PKR activation is the downstream of mt-dsRNA released by PNPT1 reduction.

7. Please show that overexpression of PNPT1 (PNPT1-AAV) successfully resulted in mitochondrial expression of PNPT1 protein.

Response: WB analysis and immunofluorescence co-labeling of mitochondrial marker and PNPT1 showed that PNPT1 AAV resulted in mitochondrial expression of PNPT1 protein compared to Ctrl AAV (new Extended Data, figure 5).

8. C16 can also inhibit other kinases including CDKs. Please confirm that knockdown of PKR results in the same phenotypes as those of C16.

Response: According to reviewer's comment, we monitored the total protein synthesis and cell apoptosis in PNPT1-deficient HK2 cells treated with C16 or PKR siRNA, respectively. As shown in new Extended Data figure 6a and 6b, knockdown of PKR indeed resulted in the same phenotypes as those of C16 in terms of protein synthesis and cell apoptosis.

9. Although the authors showed that inhibition of PKR using C16 rescued renal tubular injuries, most direct evidence will be suppressing mtRNA expression directly using IMT1 (Bonekamp et al., Nature 2020). Please analyze the kidney tissues of IRI and UO mice after administering them with IMT1.

Response: We appreciate reviewer's constructive suggestion. To directly suppress mtRNA expression, we treated HK2 cells with inositol 4-methyltransferase (IMT1), a small-molecule inhibitor recently reported by Bonekamp et al. (Nature 2020, Ref#47). As shown in new Extended Data figure 7, IMT1 treatment markedly attenuated renal tubular damage induced by IRI and UO procedure. Based on the result, the manuscript has been revised accordingly (Page 12, marked in red).

Minor comments:

1) Line 69: Please remove “as”.

Response: We have removed ‘as’ accordingly.

2) Line 69-70: Most endogenous dsRNAs are derived from Alu elements and thus are not predominantly confined to the mitochondria.

Response: We thank reviewer for pointing out this, and have removed such sentence throughout the text.

3) Line 122: Please remove “degree”.

Response: We have removed ‘degree’ in the revision.

4) Line 126: Again, endogenous dsRNAs are not predominantly derived from mitochondria.

Response: We have deleted such sentence in our revision.

5) Line 145: A statement that PNPT1 expression can be modulated by IFN needs a reference.

Response: We thank reviewer for pointing out this. Accordingly, a new reference for IFN modulating PNPT1 was cited in our revision (Sarkar et al., Cell Death Differ 2006. Ref#37).

Reviewer 2#

Major comments:

1. A major goal is to characterize pathophysiologic processes that regulate tubular atrophy, which is generally considered to be inexorable and accompanied by interstitial fibrosis, as part of chronic kidney disease progression. However, most of the data are in acute (and potentially reversible) models (especially ATN from IRI), which are not usually characterized by tubular atrophy. In the Figure 1c images, with the possible exception of DN, the interstitium does not appear to be increased (though a counterstain for scar matrix would be helpful), and tubules appear intact, suggesting absence of tubular atrophy. It might make more sense to frame the project in terms of acute tubular injury, and delete the association with tubular atrophy or chronic kidney disease.

Response: We agree with reviewer that it makes more sense to change ‘atrophy’ to ‘injury’. Based on reviewer’s suggestion, we re-framed the project in terms of acute tubular injury and changed the renal atrophy to injury through the text (The changed parts were marked in red).

2. The UUO model would be expected to cause injury predominately to collecting ducts and distal tubules, whereas AQP1 labels proximal tubules. A related issue is that mice with Cre under the control of a GGT promoter, crossed with floxed mice, would result in proximal tubule-specific knockout mice, i.e., not all tubules. Some explanation is required regarding how UUO effects would be reversed in these mice with injury to non-proximal tubules. A separate issue with the UUO model is that it is customary to ligate the ureter for a finite time period, then release the ligation, and examine residual effects. In the Methods (lines 377-378) the ureter is ligated and cut, which results in irreversible obstruction. It is difficult to envision how any intervention would rescue the phenotype of a permanently obstructed kidney.

Response: We understand reviewer’s concern on these issues. First, the UUO model used in the present study causes injury to both distal renal tubule and proximal tubule. In fact, the proximal tubule is highly vulnerable to acute injury, including ischemic insult and nephrotoxins, and chronic kidney injury. Injury of proximal tubule may result in damage of whole renal tubules (distal tubules and collecting ducts) through secretion of inflammatory cytokines and EMT. Employing UUO model, Jang et al. established renal proximal tubule injury as a primary cause of the development of chronic kidney disease (Jang et al., AJP 2021. Ref#26). Dong’s group also recently showed that UUO-induced renal fibrosis could be attenuated by deleting HIF-1 α in the proximal tubules (Mei et al., FASEB J 2022. Ref#27). Our finding of proximal tubular specific PNPT1 knockout aggravating renal fibrosis is in line with these previous reports. Second, different from the method reviewer mentioned (‘ligate the ureter for a finite time period, then release the ligation’), UUO model in which the ureter is ligated and cut is a classic renal tubular fibrosis model widely used in the field (Klahr et al., AJP 2002. Ref#28; Wu et al., KI 2006. Ref#29; Chi et al., JASN 2017. Ref#30). Such model was extensively reviewed by Saulo Klahr and Jeremiah Morrissey (AJP 2002. Ref#28), in which they highlighted the molecular and cellular pathways that culminate in renal fibrosis. Indeed, permanent kidney obstruction may change kidney function. However, instead of monitoring the kidney function loss or recovery, our study, like many others, only focused on renal tubular fibrosis during 1-2 weeks of obstruction. Employing this method, Wu et al. reported that rapamycin attenuated UUO-induced renal fibrosis (KI 2006. Ref#29). Chi et al. showed that IL-36 signaling contributed to the pathogenesis of renal tubulointerstitial lesions through the activation of the NLRP3

inflammasome and IL-23/IL-17 axis (JASN 2017. Ref #30). We have cited these works in the revision (Page 6, marked in red).

3. The use of PI in the apoptosis assays confounds the interpretation for several reasons. First, it is generally used as a marker for necrosis, or perhaps late apoptosis in PI-positive + annexin V-positive cells. More importantly, PI can yield false-positive readings due to labeling of cytosolic RNA (PMID: 20381494). A quick solution would be to merely count the annexin V-positive, PI-negative cells. Alternatively, it may be necessary to pre-treat cells with RNAses or choose a completely different apoptosis assay.

Response: According to reviewer's constructive comment, flow cytometry analysis of annexin V-positive, PI-negative cells was performed. As shown in new Fig. 4, b and e, PNPT1 knockdown markedly enhanced cell apoptosis in a time-dependent manner, whereas C16 treatment decreased the HK2 cell apoptosis induced by PNPT1 knockdown.

4. Fanconi syndrome is defined by enhanced urinary excretion of glucose, phosphorus, bicarbonate, uric acid, potassium and amino acids. These analytes should be measured to justify the claim that PNPT1 deletion phenocopies Fanconi syndrome. Although mild hypocalcemia may be observed with Fanconi syndrome, osteopenia is generally associated with hypophosphatemia. If proximal tubule deletion of PNPT1 does indeed cause Fanconi syndrome, that is an interesting finding. However, this is not a feature of IRI-induced ATN of urinary tract obstruction. To exploit the PNPT1 knockout mice in a manner that is relevant to ATN and UUO, it might be more useful to characterize the PNPT1 heterozygotes.

Response: We appreciate reviewer's insights on these critical issues. As for Fanconi syndrome, we do not claim that proximal renal tubular PNPT1 deletion causes Fanconi syndrome. All our data suggest that proximal renal tubular PNPT1 deletion markedly impairs proximal renal tubular, particularly re-absorption, leading to Fanconi syndrome-like phenotype. We clarified this in our revision. Nevertheless, we analyzed other key indicators of Fanconi syndrome according to reviewer's comment. As shown in new Fig. 7, f and g, PNPT1 KO mice displayed osteochondrosis, glycosuria, amino acid urine, and time-dependent hyperphosphaturia.

According to reviewer's comment, we also analyzed the phenotype of PNPT1 heterozygotes (Het). As shown in Figure below, PNPT1 heterozygotes displayed certain degree of renal tubular injury compared to WT littermates. However, their injury degree was markedly less than that of PNPT1-KO mice. In specific, PNPT1 heterozygotes had only mild bone loss and no bone structure deformation like that in rickets. Compared to WT littermates, PNPT1 heterozygotes displayed no or 'mild' decrease of serum calcium and phosphorus, as well as 'mild' increase of urine glucose, uric acid and urine phosphorus.

Figure 1. Phenotype analysis of PNPT1 heterozygotes. **a**, WB analysis of PNPT1 level in renal tubules from WT littermates, PNPT1 heterozygotes (Het) and tubular-specific PNPT1-deficient (KO) mice. **b**, H&E staining and TUNEL assay of renal tubular injury and apoptosis. **c**, Bone density scanning of WT, Het and KO mice. **d-e**, Serum calcium and phosphorus levels (d), and urine glucose, uric acid, urine phosphorus and potassium in WT, Het and KO mice. Data from three independent experiments were presented as mean \pm SEM, and *P* value was analyzed by ANOVA with Tukey-Kramer test. ns, $P \geq 0.05$, * $P < 0.05$, ** $P < 0.01$.

5. There are several key assays, for which the criteria/scoring system were not defined. The first is tubular injury (Figures 1 and 6). The associated box plots corresponding to tubular atrophy in Figures 1d and 1f were also not defined. For Figure 1 and 5 dsRNA immunohistochemical quantification, how was the threshold established for positive vs. negative? And is the denominator for relative staining all tubules, tubular epithelial cells, proximal tubule cells?

Response: We apologize for the missing information. Tubular injury (Fig. 1 and Fig. 6) in patients were evaluated from clinical pathological data. In mouse renal tubular injury models, tubular injury was evaluated by counting the mean number of injured tubules among total tubules in the field of view. For instance, tubular injuries in Fig. 1 and Fig. 6 were evaluated by counting the proportion of injured tubular areas in the field of view. For dsRNA thresholds in Fig. 1 and Fig. 5, dsRNA/AQP1 pixel ratio=0 was the negative threshold; dsRNA/AQP1 pixel ratio=1 was the positive threshold.

Minor comments:

1. Modest editing is required for English.

Response: The manuscript has been revised by English Professional Agency.

2. Suspect the glomerulus and tubule labels for Extended data 2b western blots are reversed.

Response: We apologize for this error and have fixed it in new Extended Data figure 4b.

3. The number of samples used to generate Figure 2b is not stated.

Response: We have stated the number of samples used in new Fig. 2b as well as in the figure legend: ‘Renal tissue samples were from 85 patients including ATN (n=12, 5 males and 7 females), DN (n=14, 8 males and 6 females), IgAN (n=16, 9 males and 7 females), LN (n=15, 5 males and 10 females), MN (n=13, 8 males and 5 females) and FSGS (n=15, 6 males and 9 females), as well as 5 non-renal tubular injury controls (3 males and 2 females). Images of 10 randomly selected regions per sample were obtained and analyzed’.

4. For detection of cytosolic mt-dsRNAs, verification that lysates are free of mitochondrial contamination should be shown.

Response: We performed western blot to analyze the purity of cytosolic lysate. As shown in new extended data figure 3, cytosolic lysates were free of mitochondrial contamination.

5. Although Figures 5a and 5e demonstrated PNPT1 AAV-injected mice do not have decreased PNPT1 expression after IRI or UUO, suggesting that the injection resulted in PNPT1 overexpression, this should be verified with western blot lanes corresponding to Sham + AAV PNPT1.

Response: Both Western blot analysis and immunofluorescence labeling showed that PNPT1 AAV successfully increased PNPT1 expression in mouse renal tubules compared to Ctrl AAV (new Extended Data figure 5).

6. Figure 7a genotyping figure requires more explanation. In particular, which lanes are WT and which are KO, and what is the significance of the boxed lanes? What are the expected band patterns for each genotype? In addition, please state whether there is any phenotype in the GGT-Cre only and floxed only mice.

Response: In Fig. 7a, lane information for genotype has been re-labeled. The expected band of WT was at 324bp (with GGT-cre negative), and KO at 428bp (with GGT-cre positive), respectively. We also monitored the phenotype of the GGT-Cre only and floxed only mice according to reviewer’s comment. As shown in new Extended Data figure 8, the GGT-Cre only or floxed only genetic modification had no effect on renal tubular morphology and PNPT1 expression (Page 12, marked in red).

REVIEWERS' COMMENTS

Reviewer #1 (Remarks to the Author):

The authors have significantly improved the manuscript, as requested. Additional experimental data complement the original ones to support the authors' claims. With the changes, I recommend accepting this manuscript for publication. It is a great paper.

Reviewer #2 (Remarks to the Author):

Reviewer #2

Major comments:

1. OK
2. OK
3. Please clarify in the Methods and/or figure legend that apoptotic cells are identified from the annexin V-positive, PI-negative fractions.
4. OK
5. The definition of an injured tubule is still not provided.

Minor comments

1. There are still issues with English, which the editors can address. The one instance where the interpretation might matter is line 121. Please clarify that Scr is serum creatinine, and not some sort of ratio, as implied in the sentence.
2. OK
3. OK
4. OK
5. OK
6. OK

Point-to-point response letter

Reviewer 2# (Remarks to the Author):

Major comments:

3. Please clarify in the Methods and/or figure legend that apoptotic cells are identified from the annexin V-positive, PI-negative fractions.

Response: According to reviewer's comment, we have clarified in the Methods that apoptotic cells are identified from the annexin V-positive, PI-negative fractions (Page 19, marked in red).

5. The definition of an injured tubule is still not provided.

Response: We apologize for the missing information. To determine tubular injury, defined as loss of the brush border, compensatory tubular dilatation and detachment, tubular apoptosis and cellular casts, a semiquantitative scoring method was used. Score 0 represents injury area less than 10%, whereas 1, 2, 3 and 4 represent the injury involving 10–25, 25–50, 50–75 and >75% of the renal tubular area, respectively. At least 12 randomly chosen fields under the microscope were evaluated for each mouse tissue section and an average score was calculated. We have added this information in the Method (Page 18, marked in red).

Minor comments

1. There are still issues with English, which the editors can address. The one instance where the interpretation might matter is line 121. Please clarify that Scr is serum creatinine, and not some sort of ratio, as implied in the sentence.

Response: We thank reviewer for pointing out this. We have clarified that Scr is serum creatinine. The manuscript has been revised again.